# MNEMODYN: LEARNING RESTING STATE DYNAMICS FROM $40$K FMRI SEQUENCES

**Sourav Pal**[1]    **Viet Luong**[1]    **Hoseok Lee**[2]    **Tingting Dan**[3]    **Guorong Wu**[3]
**Richard Davidson**[1]    **Won Hwa Kim**[2]    **Vikas Singh**[1]

[1]University of Wisconsin–Madison, Madison, United States
[2]Pohang University of Science and Technology (POSTECH), Pohang, South Korea
[3]University of North Carolina at Chapel Hill, Chapel Hill, United States

{spal9, vhluong, rjdavids}@wisc.edu   {hslee0608, wonhwa}@postech.ac.kr
vsingh@biostat.wisc.edu   {Tingting_Dan, guorong_wu}@med.unc.edu

## ABSTRACT

We present a dynamical-systems based model for resting-state functional magnetic resonance imaging (rs-fMRI), trained on a dataset of roughly 40K rs-fMRI sequences covering a wide variety of public and available-by-permission datasets. While most existing proposals use transformer backbones, we utilize multi-resolution temporal modeling of the dynamics across parcellated brain regions. We show that **MnemoDyn** is compute efficient and generalizes very well across diverse populations and scanning protocols. When benchmarked against current state-of-the-art transformer-based approaches, MnemoDyn consistently delivers superior reconstruction quality. Overall, we find that with such large-scale pre-training on (non-proprietary) rs-fMRI datasets, we get a highly performant model for various downstream tasks. Our results also provide evidence of the efficacy of the model on small sample size studies which has implications for neuroimaging studies at large where resting state fMRI is a commonly acquired imaging modality.

## 1 INTRODUCTION

Understanding the latent dynamics underlying resting-state hemodynamic signals is central to applications such as surgery planning and epilepsy seizure localization, as well as advancing neuroscience more broadly (Deco et al., 2011; Friston, 2011). Modalities such as resting-state functional Magnetic resonance imaging (rs-fMRI) provide temporal signals that encode rich neural processes (Smith et al., 2013). One important goal is to model these dynamics in a manner that captures the spatial and temporal structures of these signals (Breakspear, 2004), and permits statistical group testing and predictions across subjects, institutions/sites, and protocols (Yamashita et al., 2019). Achieving this goal requires reproducible models capable of learning representations from large-scale neuroimaging data. To this end, recent developments surrounding Foundation models (Bommasani et al., 2021) are particularly relevant. For example, while such models were originally developed for natural language processing, they have been adapted to vision (Dosovitskiy et al., 2020), robotics (Brohan et al., 2022), and beyond (Radova et al., 2025; Wang et al., 2025). Common to most foundation models are attention-based architectures – most prominently, the Transformer module – which allows flexible context modeling via self-attention. Language models (Brown et al., 2020; Chowdhery et al., 2023) achieve strong performance across most natural language tasks and so, Transformer-based architectures have also been extended to sequence modeling problems, including time series data (Zhou et al., 2021; Liu et al., 2023; Chen et al., 2025).

**Rationale for alternatives.** Transformer models serve as a good starting point for building foundation models for fMRI. Recent work (Caro et al.; Dong et al., 2024) has shown that attention-based models can capture temporal dependencies in rs-fMRI sequences (Kim et al., 2023) and model long-range dependencies well. These approaches work well when sufficient computing resources and data are available, particularly for standard acquisition protocols that typically collect 5-7 minutes of resting-state data (Birn et al., 2013). But there are some reasons to explore alternatives. First, emerging

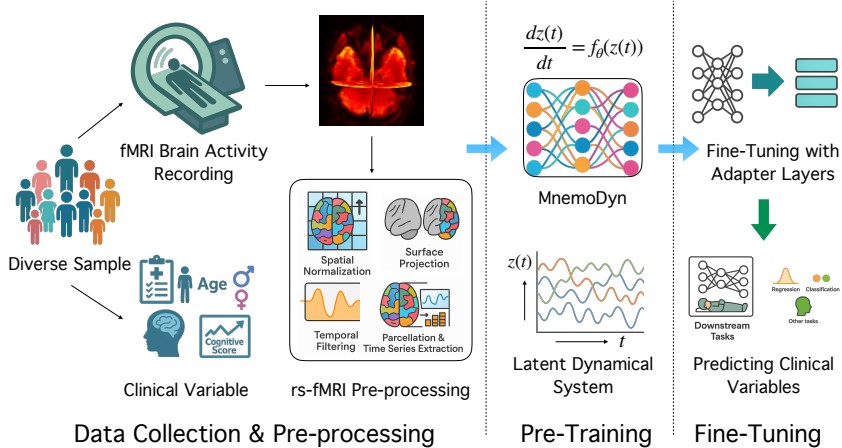

Figure 1: Overview of the **MnemoDyn** framework. Our foundation model for rs-fMRI treats temporal signals as trajectories in a latent dynamical system, parameterized by learnable operators. The pipeline begins with data curation and pre-processing of large-scale **rs-fMRI** cohorts into standardized gray-ordinate representations. The pre-training stage learns a multi-resolution non-linear dynamical operator via $\frac{dz(t)}{dt} = f_\theta((z(t))$, enabling the model to capture cross-scale temporal dependencies while preserving local dynamics. During fine-tuning, lightweight adapter layers adapt the pretrained model to diverse downstream cohorts. The resulting representations support prediction of clinical variables across heterogeneous populations (age, sex, cognitive traits, neurodegeneration markers, etc.), highlighting MnemoDyn's ability to generalize from large-scale dynamical modeling to scientifically meaningful tasks.

use cases in sleep research and clinical neuroscience (Yang et al., 2024) are increasingly utilizing longer acquisitions (e.g., up to 8 hours of continuous rs-fMRI data). Characterizing rs-fMRI dynamics from whole-night fMRI means being able to process much longer sequences without increasing the computation dramatically. Second, the data requirements for fine-tuning foundation models on *smaller* datasets, a common scenario in practice, will be simpler via more sample-efficient architectures. Third, a more compute/parameter efficient architecture will be easier to deploy in the clinic.

**Learning dynamics with operators.** In contrast to attention-based sequence modeling, time series data from neuroimaging can benefit from approaches that seeks to model the underlying dynamical structure of brain activity. Instead of learning autoregressive mappings in the raw signal space or latent space, we can attempt to learn the underlying *operator* governing the observed temporal dynamics. This casting treats the brain as if it is generating trajectories in a high-dimensional latent space governed by an unknown but learnable dynamical system. This aligns, at least partially, with neuroscientific understanding of brain activity as arising from complex dynamical processes (Deco et al., 2011; Breakspear, 2004; Sanz-Leon et al., 2015). Recent work in state space models (Gu et al.) and liquid-FM (Hasani et al.) has also highlighted the value of using dynamical system–inspired strategies for sequence modeling. Our architecture, **MnemoDyn**, learns an evolution operator rather than relying on autoregressive sequence modeling and implicitly modeling hidden state recurrence. Specifically, the solution (operator) maps initial conditions and inputs to full latent trajectories and we parameterize the evolution kernel using multi-resolution wavelet bases partly inspired by studies in neuroscience (D'Angelo & Jirsa, 2022; Basar et al., 1999). This way, the operator can process features across multiple temporal scales while maintaining temporal locality. The interaction between wavelets and pseudo-differential operators yields highly sparse representations (bey, 1991), leading to computational efficiency. The **key contributions** are:

(i) We describe a wavelet-parameterized evolution operator that captures multiscale temporal dependencies without attention mechanisms, and it scales efficiently to long sequences. This design eliminates the need for positional embeddings or tokenization schemes, which are often finicky, domain dependent, and sensitive to hyperparameter tuning.

(ii) We obtain consistent improvements over state of the art transformer-based baselines across multiple rs-fMRI datasets for reconstruction, classification, and regression tasks.

(iii) *Open-source rs-fMRI foundation model:* MnemoDyn trained on 40K rs-fMRI sequences will be publicly released for use and fine-tuning on smaller datasets.

Our findings suggest that operator-based models grounded in multi-resolution analysis offer a powerful alternative to attention-heavy architectures for modeling the dynamics associated with rs-fMRI data. They open new avenues for data-driven discovery of brain dynamics that are principled and practically deployable in resource constrained settings.

## 2 MODELING BRAIN DYNAMICS

We now set up our formulation for modeling the latent dynamics of the rs-fMRI signal. We consider the temporal hemodynamic measurements as if it is generated by a latent dynamical system defined on a low-dimensional multiscale representation of the observed neural signals. We model the evolution using a latent dynamical system (John et al., 2022) framework, i.e., decomposing the observed signal into a hidden neural process (McCormick et al., 2022) and a measurement process.

Let $x(t) : t \to R^n$ denote the observed time series data (e.g., rs-fMRI) which is high-dimensional. We use $x_t \in \mathbb{R}^n$ to denote the observed signal at the discrete time step $t$, where $n$ is the number of brain regions (e.g., parcels, voxels, or channels in case of EEG). Let $z_t \in \mathbb{R}^d$ represent the latent neural state, which is assumed to evolve according to some (potentially nonlinear) dynamics. The general form of the state-space model (Gu et al.; Gu & Dao) is given by:

$$z_{t+1} = f(z_t, u_t; \theta) + w_t, \qquad \text{(latent state transition)} \qquad (1)$$
$$x_t = h(z_t; \phi) + v_t, \qquad \text{(observation model)} \qquad (2)$$

where $u_t \in \mathbb{R}^m$ is an optional exogenous input (e.g., stimulus or task), which may optionally be zero. Here, $f : \mathbb{R}^d \times \mathbb{R}^m \to \mathbb{R}^d$ defines the transition dynamics with parameters $\theta$ and $h : \mathbb{R}^d \to \mathbb{R}^n$ is function that maps the latent space to the observations with parameters $\phi$; $w_t \sim \mathcal{N}(0, Q)$, $v_t \sim \mathcal{N}(0, R)$ are potential process and observation noise, respectively.

*Remark* 2.1. A growing body of work shows how to extend such state-space models to large-scale sequence modeling (Gu et al., 2020; Gu et al.). Also related are results describing liquid foundation models (Hasani et al.) showing excellent results via input dependent adaptivity. While these approaches also learn state evolution functions, our approach is complementary in that we utilize a special multi-resolution decomposition, which is compute-efficient and also inspired by empirical studies in neuroscience.

**Benefits of continuous time formulation:** While the state-space formulation in discrete time is widely used, modeling neural dynamics in continuous time offers some advantages (Billings et al., 2017). Brain signals are inherently continuous, even though the measurements (such as fMRI) are obtained at discrete time points. By using a differential equation-based formulation, we seek to better model the temporal evolution of neural states in a continuous form. We know that ordinary differential equations (ODEs) (Hartman, 2002) provide a principled way to encode smoothness, locality in time, and known structural constraints such as linear relaxation, oscillatory components, or conservation laws that reflect biophysical priors. This continuous-time perspective opens the door to tools from control theory and operator learning for analyzing the evolution of latent neural processes (Lee et al., 2022; Cai et al., 2021). This strategy also allows us to circumvent tokenization and patching strategies required for attention based models.

**Parameterized ODEs as Continuous Time Models:** We consider the latent state $\mathbf{z}(t) \in \mathbb{R}^d$ as evolving according to an ordinary differential equation (ODE) driven by a vector field $F : \mathbb{R}^d \times \mathbb{R}^m \to \mathbb{R}^d$,

$$\frac{d\mathbf{z}(t)}{dt} = F(\mathbf{z}(t), \mathbf{u}(t); \theta), \quad \mathbf{z}(0) = \mathbf{z}_0, \qquad (3)$$

where $\mathbf{u}(t) \in \mathbb{R}^m$ denotes an optional external control or input signal, and $\theta$ are parameters governing the dynamics (Chen et al., 2018; Finlay et al., 2020). Starting from state space naturally led us to the ODE formulation. This may seem restrictive, but we will soon see the more general form that will nicely deal with non-Markovian dynamics through an integral operator. We note that while $f$ and $F$ are distinct mathematical objects: one a discrete map and the other a differential operator, they are tightly connected. $f$ approximates the time-$\Delta t$ flow of the ODE defined by $F$. Note that $f$ can often be derived from $F$ by applying a numerical integration scheme. As an example, the Euler method would yield the update:

$$\mathbf{z}_{t+1} \approx \mathbf{z}_t + \Delta t \cdot F(\mathbf{z}_t, \mathbf{u}_t), \quad \text{s.t.} \quad f(\mathbf{z}_t, \mathbf{u}_t) = \mathbf{z}_t + \Delta t \cdot F(\mathbf{z}_t, \mathbf{u}_t) \qquad (4)$$

It is well-known that the discrete-time dynamics $f$ can be viewed (in this case) as an Euler step approximation of the continuous flow generated by $F$.

*Remark* 2.2. More sophisticated numerical integrators (Dahlquist & Björck, 2012) (such as Runge–Kutta methods) yield more accurate discrete update functions $f$ by evaluating $F$ at multiple intermediate points. The continuous-time function $F$ defines a vector field that characterizes the instantaneous rate of change of the latent state $\mathbf{z}(t)$. In contrast, the discrete-time update function $f$ governs how the latent state evolves from one time step to the next.

**ODEs to Operator Learning:** The continuous time formulation treats $\mathbf{z}(\cdot)$ as a function in a suitable function space (e.g., $C^1([0,T];\mathbb{R}^d)$) and models the evolution as a map from functions to functions. Specifically, the solution $\mathbf{z}(t)$ of the ODE (3) can be viewed as the output of a parameterized nonlinear operator $\mathcal{T}_\theta$ acting on function space (Boullé & Townsend, 2024; Subedi & Tewari, 2025):

$$\mathcal{T}_\theta : (\mathbf{z}_0, \mathbf{u}(\cdot)) \mapsto \mathbf{z}(\cdot) \tag{5}$$

which maps an initial state and a control function to the full latent trajectory. $\mathcal{T}_\theta$ denotes the flow operator induced by the vector field $F$ under the initial condition $\mathbf{z}(0) = \mathbf{z}_0$ and $\theta$ denotes the parameters. The observation model in continuous time is typically given by

$$\mathbf{x}(t) = H(\mathbf{z}(t); \phi) + \mathbf{v}(t) \tag{6}$$

where $H : \mathbb{R}^d \to \mathbb{R}^n$ is the observation function with parameters $\phi$, and $\mathbf{v}(t) \sim \mathcal{N}(0, R)$ denotes continuous-time observation noise (often modeled as white noise or band-limited noise in practice). The foregoing perspective aligns naturally with the framework of **operator learning** (Kovachki et al., 2023; 2024), where the goal is to learn mappings between infinite-dimensional objects—such as functions or distributions—rather than pointwise predictors. We will now describe our choice of the operator and check the potential benefits.

**Operators for representing rs-fMRI dynamics:** We note that (3) can be expressed in integral form by integrating the vector field $F$ over time. Assuming sufficient regularity of $F$, the latent trajectory satisfies the Volterra-type integral equation (Brunner, 2017):

$$\mathbf{z}(t) = \mathbf{z}_0 + \int_0^t F(\mathbf{z}(\tau), \mathbf{u}(\tau); \theta) d\tau \tag{7}$$

which defines a non-linear integral transform governed by $F$. We emphasize the integral from $0$ through $t$ is important: at each time point, the operator $K_\theta$ has access to the entire history of the input trajectory $u(\cdot)$ from the initial time to the current time. To make the operator structure more explicit, we observe that the integral in (7) can be expressed as a kernel integral operator acting on the control function $\mathbf{u}(\cdot)$. Suppose that the vector field $F$ admits the decomposition

$$F(\mathbf{z}(t), \mathbf{u}(t); \theta) = P(\mathbf{z}(t); \theta) + K(\mathbf{z}(t); \theta) \mathbf{u}(t) \tag{8}$$

where $P : \mathbb{R}^d \to \mathbb{R}^d$ represents an autonomous drift term, and $K : \mathbb{R}^d \to \mathbb{R}^{d \times m}$ defines a control-dependent (Kidger et al., 2020) modulation of the dynamics. Substituting into (7), we obtain

$$\mathbf{z}(t) = \mathbf{z}_0 + \int_0^t P(\mathbf{z}(\tau); \theta) d\tau + \int_0^t K(\mathbf{z}(\tau); \theta) \mathbf{u}(\tau) d\tau \tag{9}$$

The final term defines a non-linear integral operator acting on $\mathbf{u}(\cdot)$, with the kernel $K(\mathbf{z}(\tau); \theta)$:

$$(\mathcal{K}_\theta \mathbf{u})(t) := \int_0^t K(\mathbf{z}(\tau); \theta) \mathbf{u}(\tau) d\tau \tag{10}$$

which allows the latent trajectory to be compactly represented as

$$\mathbf{z}(t) = \mathbf{z}_0 + \int_0^t P(\mathbf{z}(\tau); \theta) d\tau + (\mathcal{K}_\theta \mathbf{u})(t) \tag{11}$$

The above formulation emphasizes the operator-theoretic nature of the system: the trajectory $\mathbf{z}(t)$ results from the action of a parameterized nonlinear integral operator on the control input $\mathbf{u}(\cdot)$, combined with an autonomous term driven by $P$.

We can re-write (11) as:

$$z(t) = z_0 + \int_0^t P(z(\tau)), d\tau + \int_0^t K(z(\tau)) du^W(\tau), \tag{12}$$

where $u^W(\tau)$ is the multi-scale control path associated with the corresponding Controlled Differential Equation (CDE) (Kidger et al., 2020). In CDE terminology, the wavelet-transformed path serves as the "rough path" (Morrill et al., 2021) that encodes history beyond point-wise evaluation. **MnemoDyn** implements CDEs which are a generalization of ODE, and can capture non-Markovian dependencies.

*Remark* 2.3. MnemoDyn's formulation is an integral equation with a learned kernel $K$. The main design choice is parameterization of the integral kernel which avoids the use of numerical solvers otherwise prominent on Integral Equation (IE) based formulations like (Zappala et al., 2024). MnemoDyn is contextualized within the broader operator learning literature in Appendix A.5

**Multi-Resolution Kernel Parameterization:** We focus our attention on parameterizing the kernel $K(\mathbf{z}(\tau);\theta)$ in a manner that captures the multi-scale structure of neural signals discussed in empirical studies in neuroscience (Friston, 2008; Hilgetag & Goulas, 2020; Vidaurre et al., 2017; Kiebel et al., 2008). To reflect this hierarchical organization, we adopt a *multi-resolution analysis* (MRA) framework based on wavelet bases, enabling the kernel to decompose and act on the input $\mathbf{u}$ across multiple temporal resolutions. Formally, we represent the kernel as a linear combination of separable wavelet bases:

$$K(\mathbf{z}(\tau);\theta) = \sum_{j=0}^{J}\sum_{k}\phi_{j,k}(\tau)A_{j,k}(\mathbf{z}(\tau);\theta) \tag{13}$$

where $\phi_{j,k}(\tau)$ denotes the wavelet basis function (Mallat, 1999) at scale $j$ and translation $k$, and $A_{j,k}$ are matrix-valued functions parameterized by $\theta$, modulated by the current state $\mathbf{z}(\tau)$. This decomposition provides the kernel with both temporal locality and scale adaptivity, allowing the system to selectively attend to features of $\mathbf{u}(\cdot)$ at different temporal resolutions. Substituting this into the operator (10), we obtain:

$$(\mathcal{K}_{\theta}\mathbf{u})(t) = \sum_{j=0}^{J}\sum_{k}\int_{0}^{t}\phi_{j,k}(\tau)A_{j,k}(\mathbf{z}(\tau);\theta)\mathbf{u}(\tau)d\tau \tag{14}$$

The functional role of the wavelet basis is better appreciated when we rewrite the operator by regrouping terms so that the wavelets act directly on the input signal. We can interpret $\phi_{j,k}$ as localized filters applied to the control signal $\mathbf{u}(\tau)$ at multiple scales and temporal positions. Specifically, we define the modulated input:

$$\mathbf{u}_{j,k}(\tau) := \phi_{j,k}(\tau)\mathbf{u}(\tau) \tag{15}$$

and express the integral operator from (14) as:

$$(\mathcal{K}_{\theta}\mathbf{u})(t) = \sum_{j=0}^{J}\sum_{k}\int_{0}^{t}A_{j,k}(\mathbf{z}(\tau);\theta)\mathbf{u}_{j,k}(\tau)d\tau \tag{16}$$

This makes explicit the two-stage nature of the **operator**: the input signal is first filtered by wavelet functions $\phi_{j,k}$, isolating localized features at different temporal resolutions Chen et al. (2025), and then transformed by matrix-valued operators $A_{j,k}$ conditioned on the latent state $\mathbf{z}(\tau)$. This decomposition allows the system to learn dynamic responses that are both temporally localized and state-dependent. MnemoDyn evaluates the operator on wavelet-integrated summaries of the entire past trajectory, giving it a non-Markovian property. Finally, these nonlocal integrals are coupled across space (different ROIs) and time together to modulate the latent dynamics of rs-fMRI.

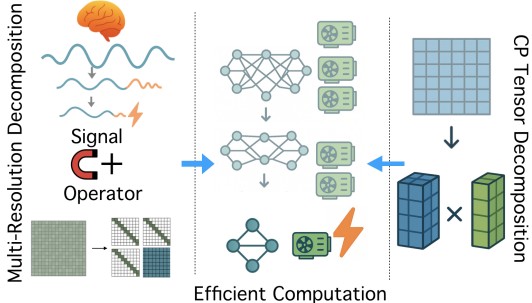

Figure 2: Wavelet-based multi-resolution decomposition of rs-fMRI signals and operator along with CP tensor factorization of model parameters enable efficient computation in **MnemoDyn**.

**Compute challenge:** While the above formulation is grounded in known priors for brain signal modeling, there are a few caveats. First, with the parameterization introduced in (16), we notice that it needs huge matrices (with increasing sequence length) and a large number of them to cover different scales and locations. Second, rs-fMRI data is inherently high dimensional, and hence we need a very large latent dimension to evolve the dynamics. These two issues would render the formulation not very useful, especially if we are targeting compute efficiency. Next, we will discuss how we circumvent these issues in practice Fig. 2.

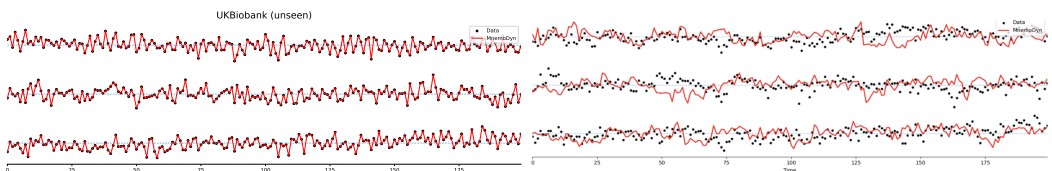

Figure 3: Reconstruction of rs-fMRI for UK-Biobank. Three parcels of unseen data are shown. We show the output of vanilla auto-encoding on the left and masked auto-encoding on the right.

**Pseudo-differential operator:** Given the inherently non-local and multiscale structure of neural signals, we note that pseudo-differential operators (Hörmander, 2007) are a potential choice for parameterization of the kernel. Interestingly, wavelets interact in a very fascinating way with pseudo differential operators leading to a sparse representation (bey, 1991). In fact, our choice of representing the signal in wavelet space in (15) and (16), makes pseudo-differential operators an especially attractive choice: (a) the interaction between the model parameters and latent dynamics is now fully expressed in the wavelet domain, and (b) the operator admits a highly compact, block diagonal representation in this basis leading to both expressive and computationally efficient modeling, as shown in (bey, 1991; Pal et al., 2023).

**Low-Rank parameter decomposition:** Due to the large number of spatial locations (voxels, sensors) dynamics of rs-fMRI, we need a high-dimensional latent space. This dimensionality poses challenges for learning and generalization, especially when the sample size is limited. To address this, we adopt a structured low-rank (Kishore Kumar & Schneider, 2017) representation of the model parameters using tensor decompositions, which provide a principled way to reduce the number of free parameters while preserving expressivity. Specifically, we employ the *Canonical Polyadic* (CP) to parameterize the collection of operators or parameter matrices in our model. The CP decomposition represents a tensor $\mathcal{X} \in \mathbb{R}^{I_1 \times I_2 \times \cdots \times I_N}$ as a sum of rank-one outer products:

$$\mathcal{X} \approx \sum_{r=1}^{R} \lambda_r a_r^{(1)} \otimes a_r^{(2)} \otimes \cdots \otimes a_r^{(N)} \tag{17}$$

where $\lambda_r \in \mathbb{R}$ and $a_r^{(n)} \in \mathbb{R}^{I_n}$ are mode-$n$ factor vectors. Each term captures a separable interaction across modes. This structure is particularly well-suited for representing the parameter tensors associated with our pseudo-differential operators in the wavelet space, enabling scalable and structured modeling of complex latent dynamics associated with brain signals.

*Remark* 2.4. We use the standard latent-space dynamical-systems framing for the problem, where BOLD activity is modeled through evolution in a lower-dimensional latent space rather than through explicit neurophysiology. **MnemoDyn** parameterizes via a multi-resolution pseudo-differential operator, which decomposes temporal interactions across scales and offers interpretability in terms of operator spectrum and scale-specific influence. This structure, however, should not be regarded as physiological validation, a limitation shared with existing data-driven baseline rs-fMRI models such as Brain-JEPA (Dong et al., 2024) and BrainLM (Caro et al.).

**Implementation details:** While the formulation above describes modeling the dynamics via a single latent space, we observe that stacking multiple such blocks, with varying latent dimensions, helps improve the representation of the model. Furthermore, adding residual connections akin to the design choice in transformer blocks further enhances performance. The block diagonal parameterization arising from the design choice of applying wavelet transforms on pseudo-differential operators (mentioned above) is implemented using multiple convolution filters and computation is parallelized for efficiency. We observe very minor variations when checking the impact of different wavelet families. We use db2 for all experimental results presented in the paper.

*Remark* 2.5. **MnemoDyn** is positioned as a *foundation model within rs-fMRI* as it exhibits clear scaling trends with model depth and pre-training data, and transfers robustly across heterogeneous datasets and downstream tasks. These are elaborated in Section 3 and in Appendix A.4.

## 3 EXPERIMENTS

Here, we describe the end to end pipeline for **MnemoDyn**. We first list the datasets used both in the pre-training and finetuning phases. Then, we describe the training procedure for both stages. Finally, we present empirical evidence showing the efficacy of MnemoDyn. A variety of additional details are included in the Appendix. Our code and pre-trained model will be publicly available.

## 3.1 DATASETS

We performed large-scale self-supervised pretraining using two different datasets. The first is **UK Biobank** (Miller et al., 2016), a public resource that provides the largest publicly available resting-state fMRI data for $\sim 65K$ samples. We consider subjects in the age range 44 to 69. The data were collected across multiple sites with a temporal resolution of 0.735s. We trained a separate model utilizing rs-fMRI data from the **Human Connectome Project (HCP)** (Human Connectome Project, 2017) with temporal resolution of 0.72s. This is a medium size dataset but has much larger sequence lengths, 1200 compared to about 500 in **UK Biobank**. The sample size is $\sim 1000$. As we will discuss later, both these models are comparable in terms of representation capability and improved performance when fine-tuned for predictive tasks. In each case, we used $80\%$ of this dataset for pretraining (including computation of normalization statistics) while the remaining $20\%$ was held out for validation purposes and internal downstream evaluations on age and sex prediction.

*Remark* 3.1. Pre-training MnemoDyn (with $92M$ parameters) can be performed on a single A100 40 GB GPU in $\sim 3$ hours with maximum memory usage. This is much cheaper compared to the configuration of 4 GPUs in other baselines (Dong et al., 2024). We attribute this to the efficient design choice of operators in MnemoDyn implemented using convolution kernels. Further, this also enables training a sample efficient foundation model utilizing data from **HCP** alone which would otherwise be insufficient for attention-based models (Caro et al.; Dong et al., 2024).

In order to demonstrate the effective representation power of MnemoDyn, we use six additional rs-fMRI datasets. The choice of these datasets is based on the two important baseline models we compare with, namely Brain-LM (Caro et al.) and Brain-JEPA (Dong et al., 2024). These datasets are as follows: (a) **HCP-Aging** (Elam et al., 2021), containing rs-fMRI from 656 elderly participants, used for trait (Neuroticism and Flanker score) and demographic (age and sex) prediction. (b) **ADNI** (Jack Jr et al., 2008), which includes rs-fMRI scans for studies of Alzheimer's disease. Following (Dong et al., 2024) we used 189 participants for normal control (NC) vs. mild cognitive impairment (MCI) classification, and 100 cognitively normal participants for amyloid positive vs. negative (c) **Healthy Brain Network** (Alexander et al., 2017) which is a large-scale pediatric classification.

neuroimaging initiative, was used for the classification of age and sex. (d) **ADHD-200** (Bellec et al., 2016), consisting of rs-fMRI from children and adolescents, used for sex classification and attention deficit/hyperactivity disorder vs. typically developing control classification. (e) **ABIDE (Autism Brain Imaging Data Exchange)** (Di Martino et al., 2014), which aggregates data across multiple sites,

| Foundation Model | UK-Biobank (MSE, $R^2$) | HCP (MSE, $R^2$) |
|---|---|---|
| MnemoDyn-UKB | 2.36e-5, 0.985 | 4.52e-08, 0.934 |
| MnemoDyn-HCP | 1.86e-09, 0.969 | 3.94e-06, 0.987 |

Table 1: Validation reconstruction MSE and $R^2$ score. We see good generalization across foundation models trained with different data.

used for autism spectrum disorder (ASD) versus control classification. (f) **NKIR** (Tobe et al., 2022), a clinical neuro-imaging resource with diverse populations, used for classification of sex.

## 3.2 PRE-PROCESSING

We now describe the pre-processing steps that were performed to convert and standardize the raw rs-fMRI data to the time-series format on which the foundation model was trained. **(a)** *NIfTI → CIFTI:* Raw BIDS-formatted rs-fMRI volumes were first converted into HCP-style dense time series (`.dtseries.nii`), aligning cortical and subcortical signals within the 91282-gray-ordinate space. This step ensures subject-level consistency and enables downstream use of surface-based atlases. **(b)** *CIFTI → Parcellation:* Each `.dtseries.nii` was then parcellated into region-level time series using the well-established atlases, yielding matrices $\mathbf{X} \in \mathbb{R}^{T \times N}$ where $T$ is the sequence length and $N$ is the number of parcels. Following (Dong et al., 2024), we used $N = 450$ using Schaefer-400 (Schaefer et al., 2018) for cortical regions and Tian-Scale (Tian et al., 2020) for sub-cortical regions. **(c)** *Normalization:* Finally, parcel-wise signals were robustly normalized based only on training data statistics, using median and interquartile range (IQR). This procedure reduces the influence of outliers and harmonizes input scales across regions and subjects. Complete algorithmic details, including projection steps, atlas handling, and normalization equations, are provided in Appendix A.1.

## 3.3 PRE-TRAINING AND FINE-TUNING

**Architecture:** Our model **MnemoDyn** were briefly outlined in Sec. 2. Each layer in MnemoDyn operates at a different wavelet scale, enabling the model to capture both fine-scale fluctuations and long-

range temporal structure in resting-state dynamics. Layers are coupled through residual connections, which progressively integrates information across scales and helps mitigate vanishing signal issues common in long sequences. Many of these layers come together to comprise an encoder block which transforms the data from one function space to another *(operator)*. MnemoDyn stacks multiple of these blocks. The input data is of sequence length 1200 (for HCP) and 490 (for UK Biobank) with $N = 450$ brain regions (ROIs). The backbone first projects these ROI signals into a hidden space, compresses them into a low-rank bottleneck. Temporal dependencies are modeled using wavelet bases (db2) stacked over multiple resolution levels. Unless otherwise specified, nonlinearities are tanh, chosen for their stability in continuous-time operator approximations.

**Pre-Training Strategy:** We evaluated several different strategies. MnemoDyn-Denoise is trained using denoising auto-encoding objective outlined in Appendix A.2.2. MnemoDyn-Mask is a *masked autoencoder* objective that masks temporal and spatial blocks at random and learns to reconstruct them from surrounding context, forcing the model to learn long-range dependencies as shown in Fig. 4 and Appendix A.2.3. We also adapt the masking scheme from (Dong et al., 2024) for MnemoDyn-Mask-JEPA. We mask 70% of the observed signal spanning temporal and spatial dimensions in the models presented here. AdamW is used as the optimizer with cosine annealing and warm restarts, gradient clipping, and early stopping. The objective is a combination of mean squared error (MSE) and mean absolute error (MAE).

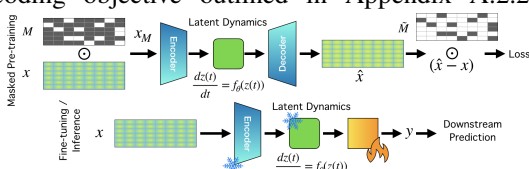

Figure 4: Masked pre-training and fine-tuning of **MnemoDyn**. We note that the dynamics are evolved in the latent space followed by the encoder, which is eventually used to fine-tune adapter layers for downstream tasks. In pre-training phase, loss is computed only on the unseen spatio-temporal data.

**Fine-Tuning:** For downstream tasks, we freeze the pretrained backbone and attach a task-specific regression/classification head which is trainable. The default head is a multi-layer perceptron (MLP) applied to pooled backbone features (averaged over time and ROIs). In particular, inputs are mapped through successive layers, each followed by LayerNorm, GELU, and dropout ($p = 0.1$), before a final linear layer outputs logits. The training objective depends on the task: mean squared error (MSE) for regression tasks (e.g., age prediction), and cross-entropy for classification tasks (e.g., sex, diagnosis). Adam is used as the optimizer with a ReduceLROnPlateau scheduler. Unless otherwise noted, robust normalization ( A.1.3) is retained while extracting features from our pre-trained MnemoDyn.

| Methods | NC/MCI (ADNI) | | Amyloid +ve/−ve (ADNI) | | Age (UKB) | Sex (UKB) | |
|---|---|---|---|---|---|---|---|
| | ACC(%) ↑ | F1(%) ↑ | ACC(%) ↑ | F1(%) ↑ | MSE ↓ | ACC(%) ↑ | F1(%) ↑ |
| BrainNetCNN | 60.00 (3.51) | 64.72 (3.18) | 59.00 (2.00) | 59.43 (1.14) | 0.99 (0.03) | 77.86 (0.98) | 78.17 (0.86) |
| BrainGNN | 67.40 (2.93) | 71.42 (2.87) | 57.00 (4.00) | 62.61 (3.48) | 0.93 (0.04) | 77.31 (0.33) | 79.23 (0.31) |
| BNT | 78.90 (4.12) | 83.14 (3.58) | 62.00 (2.45) | 59.53 (0.58) | 0.86 (0.03) | 80.78 (0.40) | 82.42 (0.36) |
| BrainLM | 75.79 (1.05) | 85.66 (1.27) | 67.00 (7.48) | 68.82 (8.48) | 0.61 (0.04) | 86.47 (0.74) | 86.84 (0.43) |
| Brain-JEPA | 76.84 (1.05) | 86.32 (0.54) | 71.00 (4.90) | 75.97 (3.93) | 0.50 (0.03) | **88.17 (0.06)** | **88.58 (0.11)** |
| MnemoDyn-Mask | **96.12 (0.31)** | **95.98 (0.29)** | **95.27 (0.39)** | **95.61 (0.37)** | 0.44 (0.05) | **88.40 (0.32)** | 88.27 (0.41) |
| MnemoDyn-Mask-JEPA | 93.67 (0.89) | 93.32 (1.09) | 94.89 (1.09) | 94.60 (1.23) | **0.42 (0.03)** | 88.30 (0.36) | 88.28 (0.41) |

Table 2: Test set performance in terms of mean (standard deviation) of MnemoDyn when fine-tuned on separate predictive tasks. Results include disease diagnosis (NC vs. MCI) and biomarker status prediction (Amyloid +ve/−ve) from ADNI (left), as well as age regression and sex classification from UK Biobank (right). Different fine-tuned variants of MnemoDyn consistently achieve lower error and higher/comparable accuracy/F1 compared to prior baselines. We report F1 score in addition to accuracy to take into account class imbalance.

## 3.4 EVALUATION

We evaluate MnemoDyn for its representation capability as well as fine-tuned performance on multiple benchmarks. Here, we present results for UK-Biobank, HCP, HCP-Aging and ADNI. Additional results on a broad basket of datasets are in the Appendix. Our choice of baselines follow (Caro et al.; Dong et al., 2024) and include existing stand-alone deep learning models (**BrainCNN** (Kawahara et al., 2017), **BrainGNN** (Li et al., 2021), **BNT** Kan et al. (2022)) as well as foundation models (**BrainLM** (Caro et al.), **Brain-JEPA** (Dong et al., 2024)) for rs-fMRI data analysis. We used graph-based and CNN

| Methods | Age | Sex | | Neuroticism | Flanker |
|---|---|---|---|---|---|
| | MSE ↓ | ACC (%) ↑ | F1 (%) ↑ | MSE ↓ | MSE ↓ |
| BrainLM | 1.14 (0.22) | 75.27 (1.24) | 73.19 (1.12) | 1.05 (0.12) | 0.77 (0.11) |
| Brain-JEPA | 1.02 (0.01) | 79.17 (1.29) | 76.29 (1.17) | 0.99 (0.01) | 1.28 (0.01) |
| MnemoDyn-Denoise | **0.91 (0.02)** | 80.2 (0.12) | 80.11 (0.11) | **0.91 (0.02)** | **0.61 (0.02)** |
| MnemoDyn-Mask | **0.90 (0.01)** | **83.10 (0.57)** | **82.77 (0.54)** | **0.90 (0.03)** | **0.60 (0.01)** |
| MnemoDyn-Mask-JEPA | **0.90 (0.01)** | 82.57 (0.35) | 82.23(0.41) | **0.90 (0.02)** | **0.60 (0.01)** |

Table 3: Performance of MnemoDyn when fine-tuned on External tasks of demographics and trait prediction on HCP-Aging. We report mean(standard deviation) for each metric. Our model achieves strong improvements in predicting age, sex, and cognitive/behavioral traits (Neuroticism, Flanker) on the test set. We report F1 score in addition to accuracy account for class imbalance.

models as our simple baselines for whole-brain rs-fMRI. More reduced models require task-specific network extraction or coarse connectivity features, which under-perform the baselines reported here.

**Representation capability of MnemoDyn:** We demonstrate the representation capacity of MnemoDyn by evaluating the reconstruction performance on held-out test data. In Fig. 3, we can see that after pre-training on UK-Biobank, MnemoDyn can faithfully recover the signal not only for unseen data. These are further numerically demonstrated in Table 1, where we perform a cross-evalutaion of MnemoDyn models trained on UK-Biobank and HCP to reconstruct held-out data from either dataset.

**Predictive Performance on Fine-Tuning:** By comparing performance of fine-tuned MnemoDyn on a variety of predictive tasks on a large collection of rs-fMRI datasets, we demonstrate the efficacy of MnemoDyn as a foundation model. We report results from two variants trained on UK-Biobank using pre-training strategies (Mask and Mask-JEPA) described above. Additional results are in Appendix.

**(i) Disease Diagnosis and Prognosis (ADNI):** We fine-tune MnemoDyn on external prediction tasks using the ADNI cohort, focusing on (a) differentiating normal controls (NC) from mild cognitive impairment (MCI), and (b) predicting amyloid biomarker status +ve/−ve (Sperling, 2011). Fine-tuned variants of MnemoDyn substantially outperform baselines (Table 2), achieving SOTA accuracy and F1 scores, underscoring its robustness and effectiveness for both diagnostic and prognostic settings.

**(ii) Demographic Prediction (UK-Biobank):** While MnemoDyn was pre-trained using UK-Biobank, we fine-tuned a model on a held-out set for predicting age and sex. We observe (Table 2) that finetuning has comparable/improved performance over baselines, underscoring the effectiveness of pre-training.

**(iii) Demographic and Trait Prediction (HCP-Aging):** We further evaluate MnemoDyn on demographic (age, sex) and cognitive/behavioral trait prediction (neuroticism, flanker) using the HCP-Aging dataset. Fine-tuned variants of MnemoDyn consistently outperform strong baselines (Table 3), yielding lower MSE and higher correlation for regression tasks (age, neuroticism, flanker) as well as higher accuracy and F1 scores for sex classification. These results highlight the versatility of MnemoDyn across both categorical and continuous prediction settings, suggesting that it works well as a generalizable foundation model. Here, we also check different variants of MnemoDyn in the pre-training strategy and observe that despite varying objectives in the pre-training phase **MnemoDyn** improves performance in downstream tasks, asserting that the performance driving factor for MnemoDyn is indeed the operator-theoretic parameterization and not merely the pre-training objective.

*Summary.* These results show the effectiveness of MnemoDyn as a foundation model which can be fine-tuned for a wide-variety of tasks. We also observe that both pre-training strategies perform equally well suggesting the robustness of MnemoDyn to small changes in the training algorithm.

**Discussion.** Our analysis shows that pre-training induces a clear multi-scale organization aligned with our wavelet-domain parameterization. Frobenius norms concentrate in operator kernels. This indicates that the dynamics are driven by structured cross-scale filters rather than dense mappings. Dense components shrink rapidly with depth, suggesting later layers rely increasingly on the global structure encoded by the operator. Sparsity patterns further support this: the wavelet-operator kernels remain fully dense (as expected for smooth, global transforms), while output-projection matrix is ∼95% sparse. Additional details in Appendix A.3.

Our code is available at https://github.com/vsingh-group/mnemodyn.

## 4 RELATED WORK

**Operator Learning and State Space Models:** Operator learning (Kovachki et al., 2023) has recently emerged as a promising paradigm for learning mappings between infinite-dimensional function spaces, particularly for modeling dynamics governed by differential equations. Early works such as DeepONet (Lu et al., 2021) and the Fourier Neural Operator (FNO) (Li et al.) demonstrated the feasibility of learning solution operators to parametric PDEs. These approaches have since been extended to irregular domains (Li et al., 2020), stochastic dynamics, and implicit formulations. State space models (SSMs) explicitly factorize latent evolution and observation mechanisms (Gu et al.). Recent works have focused on combining neural networks with structured transition kernels (Mehta et al.; Gu et al., 2020). However, majority of existing SSMs are not tailored to the specific properties of neurophysiological data, nor have they explored multiscale bases such as wavelets for learning dynamics.

**Attention-Based Models for Brain Imaging Data:** Transformer architectures have been proposed for modeling spatiotemporal data in neuroscience, including fMRI (Kan et al., 2022), EEG (Song et al., 2021), and MEG (Xu et al., 2025). Recent critical evaluations have pointed out several limitations. First, attention-based models often underperform in scenarios involving long-range, noisy, or irregularly sampled signals (Zeng et al., 2023). Second, their inductive biases are often mismatched to brain dynamics, which exhibit strong local temporal correlations, hierarchical structure, and subject-specific variability. Finally, their memory and compute requirements make them impractical for resource-constrained settings. While some transformer variants introduce inductive structure (e.g., temporal sparsity (Gorbett et al., 2023)), their complexity often remains high compared to classical or recurrent baselines. Finally, attention based models involve tokenization and patching which is a mismatch for brain imaging data derived from inherent continuous processes. Our work offers a domain-aligned alternative, avoiding global attention mechanisms entirely while still capturing long-range structure.

**Lightweight Models for Domain-Specific Sequence Modeling:** Recent works advocate against the overuse of large attention-based or foundation models for structured domains (Xu et al.) such as time series and brain imaging data. Benchmark studies (Zeng et al., 2023; Tan et al., 2024) show that compact, domain-aligned models relying on CNNs/RNNs can outperform Transformers on time series tasks, especially in low-data regimes. Recent evidence also suggests that Transformers can be over-parameterized and ill-suited for biomedical time series, where long sequences, noise, and multiscale localized dynamics dominate (Xu et al.; Kim et al.). In biomedical signal processing, convolutional and recurrent architectures have been preferred for their efficiency and robustness (Kiranyaz et al., 2015; Faust et al., 2018). In neuroscience, studies show shallow or linear models perform competitively on fMRI decoding and EEG tasks (Makeig et al., 2004), while offering better interpretability and less risk of overfitting (Haufe et al., 2014). These findings collectively motivate structure-aware architectures in domains where data is noisy, multiscale (Vidaurre et al., 2017) and expensive to acquire.

## 5 CONCLUSIONS

Domain-specific inductive biases and compact architectures often surpass large generic foundation models when data is limited or structured. This is especially true in neuroscience, where brain imaging data are noisy, heterogeneous, and expensive to acquire. Moreover, neuroimaging signals emerge from intrinsic dynamical systems with rich multiscale structure, favoring models that emphasize temporal locality and sparsity. Our work proposed a compact dynamical system based model MnemoDyn tailored for rs-fMRI. Unlike attention-based approaches, we frame learning as the identification of governing dynamical laws. We utilize the interaction between wavelets and pseudo-differential operators, and so MnemoDyn achieves efficiency/sparsity through structured decomposition, maintaining the temporal locality essential for modeling brain dynamics.

While our results of fine-tuning MnemoDyn demonstrate strong performance across diagnostic, prognostic, and trait prediction tasks, there are some limitations. First, our experiments are limited to parcellated rs-fMRI data; extending the model to voxel-level or multimodal inputs (e.g., EEG, PET) is an important future direction. Second, extending the framework to longitudinal studies will be an important next step in establishing the full potential of MnemoDyn. Addressing these limitations will help translate compact dynamical system models into robust, domain-adapted foundation models for neuroscience. While MnemoDyn provides interpretable multi-scale temporal structure, establishing genuine neurophysiological correspondence remains open and is a natural direction for future work.

## 6 REPRODUCIBILITY STATEMENT

Details of model architecture, training, and evaluation are provided in the main text and appendix. Code, pre-processing scripts, and pre-trained model weights will be released publicly via Hugging Face upon acceptance to ensure full reproducibility.

## 7 ETHICS STATEMENT

This work uses publicly available, de-identified rs-fMRI datasets under appropriate data usage agreements. No new human or animal data were collected, and the study complies with institutional and ICLR ethical guidelines. Our methods are designed for scientific research in neuroscience and health-care, and we do not foresee any direct negative societal impact. Nonetheless, as with any machine learning model applied to health-care data, risks include potential misuse for inappropriate prediction or diagnosis outside clinically validated settings. We stress that our approach is a research contribution and should not be directly applied for medical decision-making without further validation.

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

## A  APPENDIX

### A.1  PRE-PROCESSING PIPELINE

#### A.1.1  NIfTI $\rightarrow$ CIFTI `DTSERIES`.

We converted raw BIDS-formatted fMRI volumes (ADHD-200) into HCP-style dense time series (dtseries) to enable analysis in a standardized grayordinate framework. For each subject/session, volumetric BOLD runs were first mapped to cortical surfaces using ribbon-constrained volume-to-surface projection onto the `fs_LR` 32k left/right midthickness meshes, with corresponding white and pial surfaces providing anatomical constraints. Subcortical structures were handled separately by resampling the functional volumes to the `Atlas_ROIs.2` subcortical grid. These surface and subcortical representations were then integrated using the HCP 91,282-grayordinate template, producing `.dtseries.nii` files that preserve the full temporal resolution of each scan while aligning all cortical and subcortical signals to a common anatomical space. This standardized conversion ensures subject-level consistency and facilitates direct comparability with HCP-derived methods and parcellations.

#### A.1.2  CIFTI `DTSERIES` $\rightarrow$ PARCELLATION

All fMRI dense time series (`.dtseries.nii`) were parcellated into $N$ regions of interest (ROIs) using one of three brain atlases: the Gordon atlas ($N=333$; Gordon et al., 2014), the Schaefer atlas ($N=424$; Nemati et al., 2020), and the Tian atlas ($N=450$; Tian et al., 2020; Schaefer et al., 2018). Each `dtseries` is first loaded and oriented as $T \times G$ (timepoints $\times$ grayordinates), transposing when the first dimension equals $G=91,282$. Given an atlas `.dlabel.nii` with integer parcel labels over the same grayordinate space, we exclude background (0) and, for each parcel label $\ell > 0$, compute the parcel time series as the mean of all grayordinates assigned to $\ell$. This yields a parcellated matrix $\mathbf{X} \in \mathbb{R}^{T \times N}$. We verify that the atlas and `dtseries` share the same grayordinate dimension and abort otherwise.

### A.1.3 Normalization

Normalization constants are estimated *solely on the training split*. Let $X^{(i)} \in \mathbb{R}^{T \times D}$ denote the $i$-th training sample ($T = 1200$ by default), and let

$$X = \begin{bmatrix} X^{(1)} \\ X^{(2)} \\ \vdots \\ X^{(N)} \end{bmatrix} \in \mathbb{R}^{(N \cdot T) \times D}$$

be the concatenation of all training samples across subjects and time. For each ROI/feature $r \in \{1,...,D\}$, define the empirical distribution

$$\mathcal{X}_r = \{X_{t,r} \mid t = 1,...,N \cdot T\}.$$

From $\mathcal{X}_r$ we compute:

$$\begin{aligned} \text{median}_r &= \text{quantile}(\mathcal{X}_r, 0.5), \\ Q_{25,r} &= \text{quantile}(\mathcal{X}_r, 0.25), \\ Q_{75,r} &= \text{quantile}(\mathcal{X}_r, 0.75), \\ \text{IQR}_r &= Q_{75,r} - Q_{25,r}, \\ Q_{99,r} &= \text{quantile}(\mathcal{X}_r, 0.99). \end{aligned}$$

The robust scaler retains $\{\text{median}_r, \text{IQR}_r\}_{r=1}^{D}$ as fitted statistics. At transform time, a new sample $x \in \mathbb{R}^{T \times D}$ is normalized ROI-wise as

$$\tilde{x}_{t,r} = \frac{x_{t,r} - \text{median}_r}{\text{IQR}_r + \varepsilon}, \qquad \varepsilon = 10^{-6},$$

for all $t \in \{1,...,T\}, r \in \{1,...,D\}$.

## A.2 Extended Training Details

### A.2.1 Foundational training

**Backbone:** We employ *MnemoDyn*, a stacked foundation model based on wavelet multiresolution analysis. The backbone is composed of $L = 4$ blocks, each operating at a different latent dimension and coupled through residual refinement. Inputs of dimension $N = 450$ are projected to a hidden space of size 150, compressed to a low-rank bottleneck of dimensionality 5. Temporal structure is modeled with $n_{\text{levels}} = 6$ wavelet (db2) decompositions. Where required interpolation in continuous-time operators is spline-based, and nonlinearities are tanh.

**Training:** We pre-train for 50 epochs using the AdamW optimizer with learning rate $10^{-3}$, weight decay 0.01, and $(\beta_1, \beta_2) = (0.9, 0.95)$. A cosine annealing schedule with warm restarts ($T_0 = 10$, $T_{\text{mult}} = 2$) reduces the learning rate, with $\eta_{\min} = 10^{-5}$. Batch sizes are fixed at 8/16/16 for training/validation/test, respectively. The objective defaults to a composite loss, combining mean squared error (MSE) and mean absolute error (MAE). Training employs gradient clipping at 1.0, and checkpoints the two best models (lowest validation MAE) throughout training. Early stopping with patience 30 epochs is applied to prevent overfitting. Below, we describe different strategies used for self-supervised pre-training.

**Hyperparameter Tuning:** We tuned the core architectural hyperparameters that govern the capacity, multiscale structure, and locality of the operator using grid search in the range of values that are plausible based on the dimensionality of the data. The parameter controlling the hidden low-rank dimension of the operator which determines how much compression is applied during the learned integral transform was varied between $\{3,...,8\}$. The number of wavelet decomposition levels, which determine the granularity of the multiscale representation was varied between $\{4,5,6,7,8\}$ The coarse dense operator applied in each block that refines features at each scale was varied in $\{4,6,8,12\}$. The parameter that governs governs how many locally-connected filters remain active within each diagonally banded structure was varied in $\{4,6\}$.

### A.2.2 DENOISING AUTOENCODER

To encourage robustness and prevent overfitting, we employ a denoising autoencoding objective. Specifically, we corrupt the input signal by adding stochastic Gaussian noise. Given an input sequence $x$, we construct the noisy version $\tilde{x} = x + 0.1 \cdot \epsilon$, where $\epsilon \sim \mathcal{N}(0, I)$. The autoencoder is trained to reconstruct the original clean input $x$ from $\tilde{x}$. This setup forces the encoder to learn representations that are invariant to small perturbations, improving generalization.

### A.2.3 MASKED AUTOENCODER

We additionally adopt a masked autoencoding strategy. For each training sample, we randomly select five disjoint starting indices within the temporal dimension of the sequence. From each starting index, we mask a contiguous block of $80$ time steps. The model is trained to reconstruct the masked portions of the input given the unmasked context. This approach encourages the encoder to capture long-range dependencies and contextual relationships, since successful reconstruction requires leveraging both local and global sequence information.

**Temporal–Dimensional Masking.** Given a sequence of length $T$ with $D$ feature channels, we construct a binary mask $M \in \{0,1\}^{T \times D}$ by placing non-overlapping temporal windows independently for each feature. For a window length $L$ (`patch_length`) and masking ratio `mask_ratio`, the method computes

$$K_{\text{req}} = \lfloor T \cdot \texttt{mask\_ratio} \rfloor, \qquad K_{\text{cap}} = \lfloor T/L \rfloor, \tag{18}$$

and uses $K = \min(K_{\text{req}}, K_{\text{cap}})$ windows. Each feature dimension then samples $K$ disjoint start indices from $\{0, ..., T-L\}$, masking the interval $[s, s+L)$, resulting in structured temporal occlusion while keeping feature channels independent.

**Masked Reconstruction Loss.** The model is trained using a masked reconstruction objective that evaluates errors *only* on the occluded regions. Let $x \in \mathbb{R}^{T \times D}$ be the original input, $\hat{x}$ the model's reconstruction, and $M \in \{0,1\}^{T \times D}$ the binary mask. Here, $M_{t,d} = 1$ indicates a *masked* position where the loss will be computed. Before feeding the sequence into the model, all masked positions are replaced with zeros, i.e.,

$$x_{\text{masked}} = (1-M) \odot x, \tag{19}$$

so the model receives no information from the occluded intervals. The reconstruction loss is then defined as

$$\mathcal{L}_{\text{mask}} = \frac{1}{\|M\|_0} \left\| M \odot (\hat{x} - x) \right\|_2^2, \tag{20}$$

where $\odot$ denotes element-wise multiplication and $\|M\|_0$ counts the number of masked entries. By supervising the model solely on the masked regions, the learning objective forces the operator to infer the missing dynamics from surrounding temporal context, encouraging the extraction of meaningful temporal dependencies and multiscale structure rather than trivial copying from visible inputs.

### A.2.4 BRAIN-JEPA STYLE SCHEME

We also adopt the training strategy from the Brain-JEPA framework (Dong et al., 2024), which generalizes the masked prediction objective across multiple domains of the input. Concretely, given an input sequence $x \in \mathbb{R}^{T \times D}$, we define a masked subset of indices $\Omega \subseteq \{1, ..., T\} \times \{1, ..., D\}$ and construct masked views $x_{\backslash \Omega}$ by replacing entries in $\Omega$ with zeros. Masked views are sampled along three complementary axes: (i) *temporal masking*, where contiguous blocks of time steps are removed; (ii) *spatial masking*, where subsets of brain regions (ROIs) are occluded; and (iii) *cross spatio-temporal masking*, where both temporal segments and ROI subsets are jointly masked. This design compels the encoder to capture dependencies across both spatial and temporal dimensions, rather than overfitting to a single axis of variation.

Following the predictive coding paradigm of V-JEPA (Bardes et al., 2024), we train a student encoder $f_\theta$ against targets produced by a teacher encoder $f_{\theta'}$, where the teacher parameters are updated as an exponential moving average (EMA) of the student:

$$\theta' \leftarrow \tau \theta' + (1-\tau)\theta,$$

| Methods | Sex | | Age |
|---|---|---|---|
| | ACC(%) ↑ | F1(%) ↑ | MSE ↓ |
| Brain-JEPA | 58.52 (3.72) | 29.12 (1.64) | 1.02 (0.01) |
| MnemoDyn | **82.37 (2.1)** | **82.19 (1.9)** | 0.84 (0.01) |
| MnemoDyn-Mask | 81.55 (1.6) | 81.35 (1.6) | **0.80 (0.01)** |

Table 4: Performance of fine-tuned MnemoDyn on Healthy Brain Network (HBN). We report mean(standard deviation) on the test set. For both regression and classification, we get better performance than baseline.

with momentum coefficient $\tau \in [0,1)$. Given a masked view $x_{\setminus \Omega}$, the student produces predictions $\hat{x}_\Omega = f_\theta(x_{\setminus \Omega})$, while the teacher provides the reference $x_\Omega^{(\text{target})} = f_{\theta'}(x)$. The loss is the mean squared error (MSE) over the masked indices:

$$\mathcal{L}_{\text{mask}}(\theta) = \frac{1}{|\Omega|} \sum_{(t,r)\in\Omega} \left\| \hat{x}_{t,r} - x_{t,r}^{(\text{target})} \right\|_2^2.$$

By unifying temporal, spatial, and cross spatio-temporal masking, this scheme encourages the emergence of latent representations that capture both local and global structure in rs-fMRI time series.

### A.2.5 DOWNSTREAM HEAD

For downstream tasks, we freeze the pretrained backbone and attach a task-specific regression/classification head. The default head is a multi-layer perceptron (MLP) applied to pooled backbone features (averaged over time and ROIs). Concretely, inputs are mapped through successive layers, each followed by LayerNorm, GELU, and dropout ($p = 0.1$), before a final linear layer outputs logits.

The training objective depends on the task: mean squared error (MSE) for regression tasks (e.g., age prediction), and cross-entropy for classification tasks (e.g., sex, diagnosis). Optimization uses Adam ($\text{lr} = 10^{-3}$, $\text{wd} = 10^{-4}$) with a ReduceLROnPlateau scheduler (factor $0.5$, patience $5$). Batch sizes are set to $8/4$ for train/test. We use robust normalization from (Sec. A.1.3) is retained.

### A.3 ANALYSIS OF PRE-TRAINED OPERATOR

To better understand how **MnemoDyn** uses its multi-resolution operator parameterization, we decomposed the learned operator into banded, level-wise components and a small dense residual, and examined their Frobenius norms and sparsity patterns after pre-training. The results expose a highly stable multi-scale structure.

**Norm structure.** Across MnemoDynlayers, the Forbenius norm mass is concentrated in the pseudo-differential operator kernels derived from the wavelet domain. These components dominate by an order of magnitude ($\sim 1500$), indicating that the learned dynamics are primarily mediated by structured, cross-scale interactions. Dense layers exhibit rapidly decreasing norms ($\sim 20$), showing that deeper blocks rely less on local, single-scale corrections and more on the global structure encoded by the operator. Components that directly process the raw input diminish steadily with depth, consistent with hierarchical abstraction.

**Sparsity structure.** The wavelet-operator kernels remain uniformly dense across layers, which is expected for smooth, global transformations that do not benefit from element-wise pruning. In contrast, the output-projection layer is over $95\%$ sparse, revealing a highly selective readout from an otherwise rich, distributed multi-scale representation. This is consistent with the interpretation that the learned dynamics evolve on a structured, lower-dimensional manifold induced by the operator geometry.

**Activation and $L_2$ norm visualizations.** In Fig. 5 we illustrate the operator output $A_j(u_j)$ across wavelet levels $j = 0$ through $j = 4$, capturing how the learned operator responds to progressively coarser temporal representations. Level $0$ exhibits the highest temporal resolution and displays large, high-magnitude activation bursts concentrated around specific time segments, reflected by the yellow

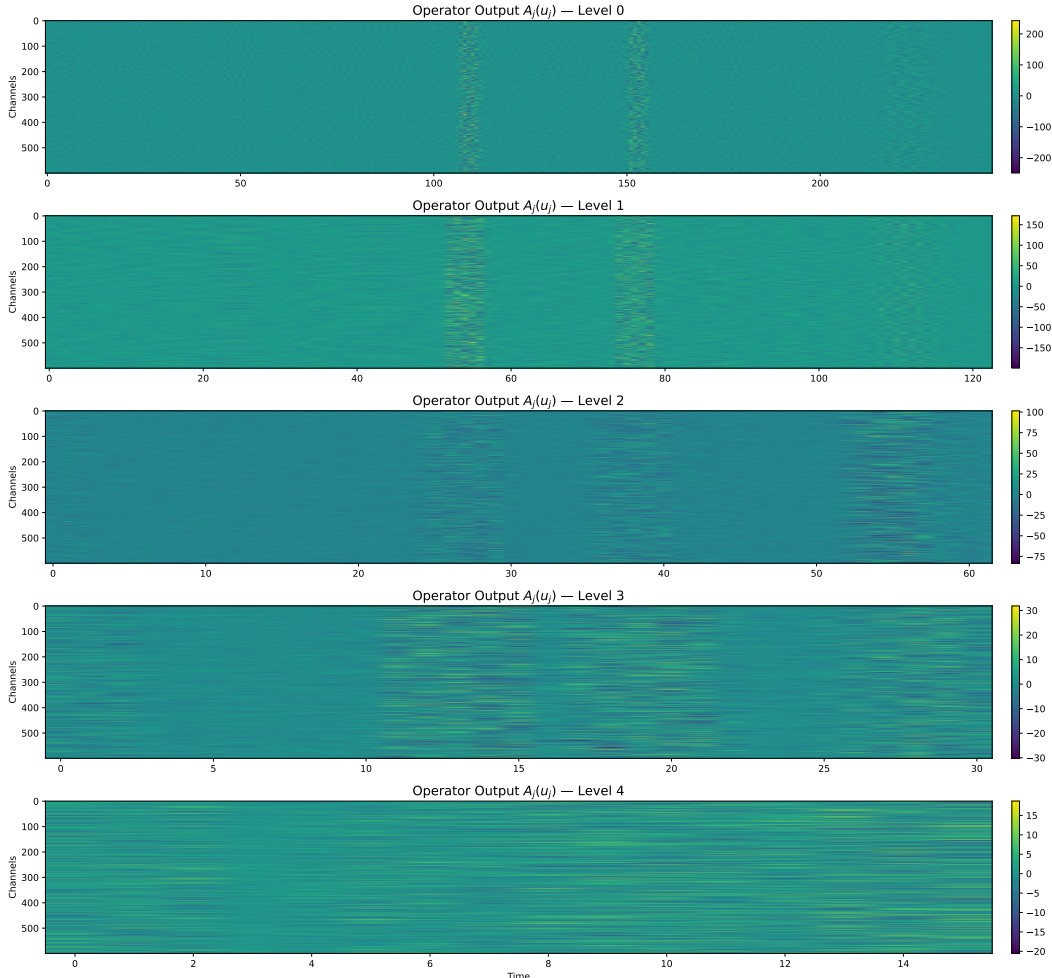

Figure 5: Operator output $A_j(u_j)$ across wavelet levels $j = 0$ to $4$.

bands. As the scale increases (levels 1 through 4), the temporal length shortens and the operator responses become smoother and notably lower in magnitude. We observe similar behavior, in Fig. 6 where we visualize the $L_2$ norm at each time step over all channels across multiple resolution(s) in the latent space of the operator output. Overall, this highlights a clear multiscale pattern: strong, localized operator activity at fine scales and progressively weaker, more diffused responses at coarser scales.

### A.4 MNEMODYN AS A FOUNDATION MODEL FOR RS-FMRI

In this section, we elaborate how **MnemoDyn** exhibits foundation-model like characteristics. We include (i) empirical evidence of scaling behavior with model size and pre-training data size, (ii) a precise delineation of generality and transfer within the rs-fMRI modality, and (iii) an explicit framing of what is and is not claimed in terms of modality breadth.

#### A.4.1 SCALING BEHAVIOR: MODEL SIZE AND DATA SIZE

We conducted controlled ablations to quantify the effect of increasing (a) the depth of the model and (b) the size of the pre-training dataset.

**Model-size scaling.** MnemoDyn consists of stacked operator blocks, analogous to depth scaling in transformer-based architectures. Increasing the number of operator blocks from *two* to *four* yields consistent improvements across regression and classification tasks on multiple downstream datasets.

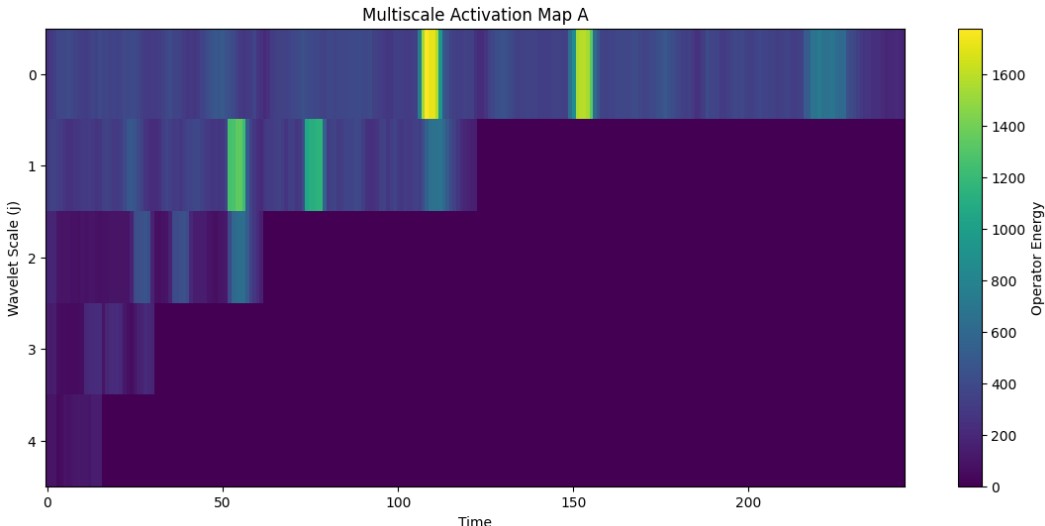

Figure 6: $L_2$ norm of operator output $A_j(u_j)$ across wavelet levels $j = 0$ to $4$.

This trend aligns with standard foundation-model behavior, where additional representational capacity improves both pre-training objectives and fine-tuning performance. The results are summarized in Table 5.

**Data-size scaling.** We additionally pre-train MnemoDyn on different subsets of UK-Biobank available subjects. Models trained with more data show uniformly stronger performance across downstream evaluations (Table 6). This confirms that MnemoDyn benefits from increased pre-training data coverage, another key requirement for foundation-model-style scaling.

The combined findings provide direct, quantitative evidence that both model depth and pre-training data size contribute to improved representational quality.

| # Blocks | Age (MSE ↓) | Sex (Acc; F1 ↑) |
|---|---|---|
| 1 | 0.914(0.012) | 0.768(0.013) 0.763(0.011) |
| 2 | 0.903(0.014) | 0.8010(0.016); 0.807(0.014) |

Table 5: Model-size scaling: downstream task performance for different numbers of operator blocks for HCP-Aging .

| # Subjects | Age (MSE ↓) | Sex (Acc; F1 ↑) |
|---|---|---|
| 1000 | 1.023(0.019) | 0.706(0.017); 0.707(0.019) |
| 10000 | 1.021(0.014) | 0.753(0.013); 0.743(0.011) |
| 20000 | 1.012(0.013) | 0.793(0.013); 0.790(0.011) |

Table 6: Data-size scaling: downstream performance when pre-trained on different fraction of UK-Biobank and finetuned for HCP-Aging.

### A.4.2 SCOPE OF TRANSFER: WITHIN-MODALITY GENERALITY AND REUSE

We clarify the exact empirical scope of generality and transfer supported by our experiments.

**Within-modality transfer (rs-fMRI).** All experiments in the present work are conducted within the rs-fMRI modality. MnemoDyn is pre-trained once and evaluated across multiple heterogeneous datasets (HBN, ADNI, ABIDE), which differ in scanner hardware, acquisition protocols, demographics, and health conditions. This cross-dataset robustness is non-trivial: each dataset required separate institutional approvals, data-use agreements, and constrained compute environments governed

by the respective data custodians. Within these practical constraints, MnemoDyn functions as a *general-purpose, reusable backbone* for rs-fMRI, consistently outperforming strong baselines across scientifically relevant tasks. This evaluation strategy is aligned with the protocols established by recent rs-fMRI foundation models such as Brain-LM (Caro et al.) and Brain-JEPA (Dong et al., 2024).

**Task generality.** Our evaluation setup of supervised regression and classification matches the dominant protocol for rs-fMRI foundation-model research. These task types enable direct, fair comparison with prior work while focusing on scientifically meaningful variables such as age, cognitive scores, and disease status. Although we do not include non-label pretext tasks (e.g., trajectory forecasting, contrastive objectives, or clustering) in this paper, MnemoDyn is architecturally designed to accommodate them: the latent trajectories $z$ produced by the model capture multi-scale temporal dependencies and spatial interactions, and can easily be used with lightweight task-specific heads for a broad family of downstream objectives.

**Cross-modality transfer.** We explicitly do *not* claim empirical cross-modality generalization here. Extending MnemoDyn to EEG or MEG requires its own elaborate process for dataset acquisition, access approval, pre-processing, and compute scheduling. While the architecture is compatible with other modalities in principle, we restrict our claims to demonstrated within-modality transfer and identify cross-modality extensions as future work.

To conclude, we define MnemoDyn as a *foundation model for rs-fMRI*, characterized by the following properties:

- **Large-scale, self-supervised pre-training** on tens of thousands of rs-fMRI sequences.
- **General-purpose, reusable representations** that support multiple downstream tasks using lightweight heads.
- **Cross-dataset robustness** across diverse scanners, sites, acquisition protocols, demographics, and health conditions within rs-fMRI.
- **Empirical scaling** in both model size and data size, as demonstrated in Tables 5 and 6.

This framing reflects the prevailing modality-specific definition used by baseline rs-fMRI foundation models.

## A.5 DISTINCTIONS BETWEEN MNEMODYN AND CLASSICAL OPERATOR-LEARNING FRAMEWORKS

Here we discuss the conceptual and architectural differences between MnemoDyn and canonical operator-learning frameworks such as the Fourier Neural Operator (FNO) and DeepONet.

**Fundamentally different problem formulations.** Classical operator-learning methods are designed for *supervised* operator regression, where the goal is to learn a map between input–output function pairs, typically arising from PDE solution operators (e.g., mapping a forcing function to its corresponding solution field). These methods assume access to explicit operator evaluations: given an input function $u$, predict the output function $\mathcal{G}(u)$, where $\mathcal{G}$ is the unknown operator to be learned.

MnemoDyn addresses a categorically different learning problem. We observe long rs-fMRI time series generated by an unknown dynamical system, with *no* paired input–output fields, no PDE solution maps, and no access to ground-truth operator evaluations. The learning objective is fully *self-supervised*, relying on reconstructing or modeling the evolution of observed sequences. While MnemoDyn contains an operator-theoretic flavor through its integral formulation of latent dynamics, the learned operator is *latent*: it parameterizes the evolution rule of the hidden trajectory rather than mapping one function to another. The kernel governs temporal modulation of the latent state, not supervised operator-valued regression.

**Architectural mismatch induced by the problem setting.** The architectural choices in FNO and DeepONet naturally follow from their reliance on paired function data, but they become ill-suited when directly applied to long, nonstationary rs-fMRI sequences.

FNO parameterizes global Fourier symbols, inducing dense global interactions across the entire domain. This is well-aligned with PDE solution operators, which are typically smooth, globally coupled mappings. However, global Fourier bases are less suitable for multi-scale, nonstationary, and locally temporal dynamics which are commonplace in rs-fMRI data. In contrast, MnemoDyn operates in a wavelet domain and parameterizes a scale-varying pseudo-differential operator whose symbol adapts across resolutions. The compact support of wavelets induces approximate banded-diagonal sparsity for long sequences, yielding both computational benefits and modeling flexibility for scale-dependent, nonstationary behavior that Fourier parameterizations do not provide.

DeepONet, on the other hand, constructs operators through two subnetworks (branch and trunk), but does not provide a clear interpretability of the resulting basis functions. This makes it difficult to justify its representational structure for multi-scale spatio-temporal brain dynamics and limits alignment with domain-specific inductive biases required for rs-fMRI modeling.

**Nature of the learned operator in MnemoDyn.**   Although MnemoDyn incorporates an operator through its integral representation of latent dynamics, it does not aim to approximate a supervised PDE solution operator. Instead, the kernel acts internally to dictate the evolution of the hidden state trajectory in continuous time. This quantity cannot be interpreted as the same object learned by FNO or DeepONet, because no input–output functional mapping is being approximated. The operator in MnemoDyn is, by construction, a latent dynamical operator extracted from unpaired, noisy observational sequences.

**Why direct comparison is non-trivial.**   Constructing a version of FNO or DeepONet suited for rs-fMRI would require a substantial redesign, including: (a) developing new objectives tailored to self-supervised sequence modeling rather than supervised operator regression; (b) replacing global Fourier bases or trunk/branch decompositions with structures that respect multi-scale temporal variability, brain parcellations, and nonstationary dynamics;(c) incorporating architectural modifications to accommodate irregular sampling and domain-specific masking used in neuroimaging; and (d) introducing neuroimaging-specific regularization schemes and inductive biases.

Such modifications amount to designing an entirely new operator-learning architecture for rs-fMRI, rather than applying an existing method unchanged. For this reason, our comparisons focus on strong, domain-appropriate rs-fMRI foundation-model baselines that align with MnemoDyn's problem setting and data assumptions.

**Summary of conceptual novelty.**   The contribution of MnemoDyn does not lie in proposing a new operator-learning formalism, but in *adapting* operator-theoretic principles to rs-fMRI in a way that is both effective and computationally efficient. The key innovations arise from the integration of (i) wavelet-domain pseudo-differential operators, (ii) structured low-rank parameterization suited to high-dimensional time series, and (iii) continuous-time evolution of latent trajectories. Taken together, these components differentiate MnemoDyn architecturally and functionally from FNO and DeepONet, whose problem formulations, assumptions, and training pipelines are fundamentally different.

blue

## A.6   IMPORTANCE OF THE MULTI-RESOLUTION ANALYSIS AND LOW RANK STRUCTURE

To assess the contribution of the wavelet-based multi-resolution analysis (MRA), we trained a version of MnemoDynin the raw latent space, removing all wavelet transforms and scale-structured operators. The "no-wavelet" variant shows a consistent degradation across all downstream tasks as shown in Table 7.

| Task | Metric | No-Wavelet | MnemoDyn-Mask |
|------|--------|------------|---------------|
| NC/MCI (ADNI) | ACC / F1 | $75.12 \pm 1.02$ / $74.39 \pm 1.01$ | $\mathbf{96.12 \pm 0.31}$ / $\mathbf{95.98 \pm 0.29}$ |
| Amyloid (ADNI) | ACC / F1 | $72.19 \pm 2.01$ / $71.23 \pm 1.78$ | $\mathbf{95.27 \pm 0.39}$ / $\mathbf{95.61 \pm 0.37}$ |
| Age (UKB) | MSE | $0.67 \pm 0.07$ | $\mathbf{0.44 \pm 0.05}$ |
| Sex (UKB) | ACC / F1 | $76.16 \pm 0.13$ / $75.14 \pm 0.23$ | $\mathbf{88.40 \pm 0.32}$ / $\mathbf{88.27 \pm 0.41}$ |

Table 7: Comparison of MnemoDyn with and without multi-resolution wavelet decomposition.

| Methods | ADHD/TDC | |
|---|---|---|
| | ACC(%) ↑ | F1(%) ↑ |
| Brain-JEPA | 45.7 (0.89) | 46.5 (0.78) |
| MnemoDyn | **54.24 (0.3)** | **54.07 (0.42)** |
| MnemoDyn-Mask | **54.36 (1.24)** | **54.26 (1.27)** |
| MnemoDyn-Mask-JEPA | **54.70 (0.75)** | **54.55 (0.80)** |

Table 8: Performance comparison on the ADHD200 dataset for ADHD vs. TDC classification. Mean accuracy (ACC) and F1-score (F1) with standard deviations are reported. Fine-tuned variants of MnemoDyn consistently outperform the Brain-JEPA baseline.

The improvements from reintroducing the wavelet-based multi-resolution structure are substantial: classification accuracy increases by **20–23 points** on ADNI tasks and by **12 points** on UKB sex prediction, while age regression error decreases by more than **34%**. These consistent gains highlight that MRA is not merely an architectural detail but a crucial inductive bias suitable for modelling the dynamics associated with rs-fMRI data.

When we remove the low-rank decomposition in the latent space, we observe immediate memory blow up for the dimensionality required to encode the latent dynamics faithfully.

### A.7 ADDITIONAL DOWNSTREAM EVALUATION ON RS-FMRI DATASETS

To assess the efficacy of **MnemoDyn** as a foundation model for brain imaging, we evaluate its fine-tuned variants across four different neuroimaging datasets spanning both regression and classification tasks. Our goal is to establish whether MnemoDyn provides consistent performance improvements over existing baselines, such as Brain-JEPA, while maintaining robustness across heterogeneous datasets. We report mean accuracy (ACC), F1-score (F1), and mean squared error (MSE), depending on the task, with standard deviations in parentheses. Across all datasets, MnemoDyn demonstrates clear advantages over the baseline, underscoring its utility as a versatile foundation model for downstream brain-related prediction tasks.

**Healthy Brain Network (HBN):**  On the HBN dataset, which includes both sex classification and age regression tasks, fine-tuned variants of MnemoDyn achieves significant performance gains as shown in Table 4. For sex classification, MnemoDyn reaches an accuracy of 82.37% and F1 of 82.19%, compared to 58.52% and 29.12% for Brain-JEPA. For age prediction, the mean squared error reduces to 0.84, outperforming the baseline error of 1.0163. The masked variant (MnemoDyn-Mask) yields similar results, further highlighting the robustness of the approach. These results indicate that MnemoDyn effectively supports both categorical and continuous prediction settings within developmental datasets.

**ADHD200:**  On the ADHD200 dataset, we evaluate the classification of Attention-Deficit/Hyperactivity Disorder (ADHD) vs. Typically Developing Controls (TDC) as shown in Table 8. Here, MnemoDyn consistently surpasses the Brain-JEPA baseline, achieving an accuracy of 54.24% and F1 of 54.07%, relative to 45.7% and 46.5% for Brain-JEPA. The masked variants further improve performance, with MnemoDyn-Mask-JEPA reaching the best scores (54.70% ACC, 54.55% $F1$). Despite the relatively challenging nature of the ADHD/TDC classification task, MnemoDyn when fine-tuned provides a clear and reproducible advantage, validating its utility across clinical cohorts.

**NKIR:**  On the NKIR dataset for sex classification, fine-tuned variants of MnemoDyn again yields substantial improvements over Brain-JEPA as shown in Table 9. While the baseline achieves 66.52% accuracy and 63.97% F1, MnemoDyn boosts performance to 87.52% for both metrics. The masked variant (88.37% ACC, 88.36% F1) offers an additional increase, demonstrating the effectiveness of the masking strategy. Overall, the NKIR results show that MnemoDyn generalizes strongly across independent cohorts, providing state-of-the-art performance on demographic prediction tasks.

| Methods | Sex | |
|---|---|---|
| | ACC(%) ↑ | F1(%) ↑ |
| Brain-JEPA | 66.52 (0.27) | 63.97 (0.40) |
| MnemoDyn | 87.52 (0.49) | 87.52 (0.48) |
| MnemoDyn-Mask | **88.37 (0.45)** | **88.36 (0.45)** |
| MnemoDyn-Mask-JEPA | 87.70 (0.29) | 87.70 (0.27) |

Table 9: Performance comparison on the NKIR dataset for sex classification. We report mean accuracy (ACC) and F1-score (F1) with standard deviations in parentheses. Finetuned MnemoDyn significantly outperform the Brain-JEPA baseline.

| Methods | Autism / TSC | |
|---|---|---|
| | ACC(%) ↑ | F1(%) ↑ |
| **MnemoDyn** | 60.32 (1.44) | 59.26 (2.04) |
| **MnemoDyn-Mask** | 58.93 (1.25) | 58.57 (1.24) |

Table 10: Classification performance on the ABIDE dataset for Autism vs. TSC for fine-tuned MnemoDyn. The table reports mean accuracy (ACC) and F1-score (F1) with standard deviations in parentheses, averaged across multiple runs.

| Methods | NC/MCI | | Amyloid +ve/−ve | |
|---|---|---|---|---|
| | ACC(%) ↑ | F1(%) ↑ | ACC(%) ↑ | F1(%) ↑ |
| BrainNetCNN | 60.00 (3.51) | 64.72 (3.18) | 59.00 (2.00) | 59.43 (1.14) |
| BrainGNN | 67.40 (2.93) | 71.42 (2.87) | 57.00 (4.00) | 62.61 (3.48) |
| BNT | 78.90 (4.12) | 83.14 (3.58) | 62.00 (2.45) | 59.53 (0.58) |
| BrainLM | 75.79 (1.05) | 85.66 (1.27) | 67.00 (7.48) | 68.82 (8.48) |
| Brain-JEPA | 76.84 (1.05) | 86.32 (0.54) | 71.00 (4.90) | 75.97 (3.93) |
| MnemoDyn-HCP-Mask | 87.34 (2.1) | 89.7 (2.2) | 88.20 (1.09) | 88.1 (1.23) |
| MnemoDyn-HCP-Mask-JEPA | **89.87 (1.09)** | **89.74 (1.0)** | **93.87 (4.05)** | **93.48 (3.07)** |

Table 11: Performance of MnemoDyn pre-trained on HCP and fine-tuned on ADNI tasks of NC/MCI diagnosis and Amyloid +ve/−ve classification. Results are reported as mean (standard deviation) for accuracy (ACC) and F1-score (F1). F1 is included to account for class imbalance across cohorts. MnemoDyn variants substantially outperform prior baselines, highlighting the benefits of pretraining on HCP before transfer to clinical tasks.

**ABIDE:** On the ABIDE dataset for Autism vs. (TSC) classification, we report results for fine-tuned variants of MnemoDyn. Due to reproducibility issues, we were unable to obtain reliable Brain-JEPA baselines on this dataset. Nevertheless, MnemoDyn achieves consistent performance, with the base model reaching 60.32% accuracy and 59.26% F1, and the masked variant obtaining 58.93% accuracy and 58.57% F1. While the margins are narrower compared to other datasets, these results underscore MnemoDyn's robustness on a highly heterogeneous clinical cohort.

## A.8 ZERO-SHOT GENERALIZATION TO SMALLER DATASETS

To further assess how **MnemoDyn** behaves in low-data regimes and to complement the main experimental results, we conducted additional analyses examining its zero-shot transfer performance. These experiments evaluate whether the pre-trained model and simple downstream heads retain predictive or reconstruction capability when applied to substantially smaller datasets without any fine-tuning.

| Methods | Age | Sex | | Neuroticism | Flanker |
|---|---|---|---|---|---|
| | MSE $\downarrow$ | ACC (%) $\uparrow$ | F1 (%) $\uparrow$ | MSE $\downarrow$ | MSE $\downarrow$ |
| BrainLM | 1.14 (0.22) | 75.27 (1.24) | 73.19 (1.12) | 1.05 (0.12) | 0.77 (0.11) |
| Brain-JEPA | 1.02 (0.01) | 79.17 (1.29) | 76.29 (1.17) | 0.99 (0.01) | 1.28 (0.01) |
| MnemoDyn-HCP-Mask | **0.90 (0.03)** | **80.90 (0.13)** | **80.70 (0.12)** | **0.90 (0.09)** | **0.58 (0.09)** |
| MnemoDyn-HCP-Mask-JEPA | **0.90 (0.04)** | **80.98 (2.13)** | **80.63 (2.12)** | **0.90 (0.57)** | **0.60(0.06)** |

Table 12: Performance of MnemoDyn pre-trained on HCP and fine-tuned on external tasks from HCP-Aging, including age regression, sex classification, and trait prediction (Neuroticism, Flanker). Results are reported as mean (standard deviation) for MSE, accuracy (ACC), and F1-score (F1). F1 is included for classification tasks to account for class imbalance. MnemoDyn variants achieve consistent improvements over prior baselines across both demographic and cognitive trait prediction.

We performed two analyses feasible within the rebuttal time frame. First, we trained simple downstream MLP heads on the larger HCP-Aging dataset and applied them directly to the dataset collected as part of the Healthy Brain Network (HBN) study. For sex classification, the zero-shot performance reached an accuracy of $0.6393$ and an F1 score of $0.6390$, while for age prediction the MLP achieved an MSE of $0.8516(0.02)$, indicating moderate transfer for categorical variables but limited transfer for age due to distributional differences.

Second, we evaluated the representation capability of pre-trained MnemoDyn on small datasets by measuring reconstruction quality without any adaptation. A model pre-trained on UK Biobank achieved an $R^2$ of $0.98$ on HBN and $0.96$ on HCP-Aging, with corresponding MSEs of $7.89 \times 10^{-8}$ and $4.52 \times 10^{-8}$, respectively. These results show that **MnemoDyn**'s learned temporal operators generalize extremely well to small datasets, maintaining high reconstruction accuracy even in the absence of fine-tuning.

Overall, these findings demonstrate that the architectural design allows the model to retain and transfer a substantial portion of its representational benefits even in zero-shot, low-data settings.

## A.9 ABLATIONS

We pre-trained **MnemoDyn** on rs-fMRI data from the HCP dataset, which contains only $\sim 1000$ subjects with sequence length 1200. Despite the modest scale of this dataset compared to conventional foundation model pretraining regimes, our operator learning formulation allows MnemoDyn to extract transferable dynamical representations that generalize strongly to downstream settings. This highlights one of the key advantages of our approach: rather than relying on massive data curation, MnemoDyn leverages inductive biases from multi-resolution operator learning to capture the underlying structure of brain dynamics in a highly data-efficient manner.

Table 11 reports results on ADNI, where we evaluate clinical tasks such as NC/MCI diagnosis and amyloid $+$ve$/-$ve prediction. Fine-tuned variants of MnemoDyn substantially outperform a wide range of prior baselines. In particular, MnemoDyn-HCP-Mask-JEPA achieves up to $89.9\%$ accuracy and $89.7\%$ F1 on NC/MCI classification, and $93.9\%$ accuracy and $93.5\% F1$ on amyloid prediction, representing a sizable improvement over all baselines. These results underscore the benefits of pretraining on HCP, even with limited sample size, before transfer to downstream clinical cohorts.

We further assess transfer to external demographic and cognitive prediction tasks from HCP-Aging (Table 12). Here, MnemoDyn consistently surpasses prior methods on age regression, sex classification, and trait prediction tasks (neuroticism and flanker). Importantly, these gains are observed across both demographic and cognitive dimensions, illustrating the broad generalization capacity of MnemoDyn.

Taken together, these ablations highlight that **MnemoDyn** can be effectively pretrained on a relatively small dataset such as HCP and still transfer with strong performance to diverse downstream tasks. This data efficiency is a direct consequence of the operator learning setup, which allows the model to learn evolution rules over brain dynamics rather than memorizing dataset-specific patterns. As

such, MnemoDyn offers a principled path toward building foundation models for neuroscience without requiring prohibitively large pretraining corpora.

## ACKNOWLEDGMENTS

Partial support for this work came from a contract to UW-Madison under the DARPA Strengthen program. We thank Tammi Kral for suggestions regarding pre-processing steps in the initial phase of the project.

