# OpenReview forum: "MnemoDyn: Learning Resting State Dynamics from  $40$K FMRI sequences"
_ICLR.cc/2026/Conference — ICLR 2026 Poster_

### Official Review · Reviewer_sskM · 2025-10-29

**Soundness:** 3
**Presentation:** 3
**Contribution:** 3
**Rating:** 6
**Confidence:** 3

**Summary:**

The paper MnemoDyn: Learning Resting-State Dynamics from 40K fMRI Sequences introduces MnemoDyn, a foundation model for resting-state fMRI that replaces attention mechanisms with operator-based dynamical system modeling. Instead of tokenized sequence learning, MnemoDyn learns multi-resolution temporal operators using wavelet parameterization and pseudo-differential kernels to efficiently capture multiscale neural dynamics. The architecture offers computational efficiency and scalability to long sequences, achieving strong reconstruction and predictive performance across datasets such as UK Biobank, HCP, and ADNI. Fine-tuned versions of MnemoDyn show improved accuracy and robustness on downstream tasks including age, sex, and disease classification. Overall, the paper argues for a biologically grounded, operator-based alternative to transformer models for neuroimaging, highlighting interpretability and compute efficiency but offering only moderate conceptual novelty and limited theoretical validation.

**Strengths:**

1. Proposes an elegant operator-based formulation for modeling brain dynamics, avoiding attention mechanisms.
2. Achieves state-of-the-art results on multiple large and small-scale fMRI datasets with high compute efficiency.
3. Presents a comprehensive evaluation pipeline, supporting claims of generalization and scalability.
4. Aligns with neuroscientific priors regarding multiscale temporal structure, enhancing biological plausibility.

**Weaknesses:**

1. Conceptual novelty is modest; builds on existing operator learning and neural ODE frameworks without substantial innovation.

2. The connection to neuroscientific principles is largely rhetorical, lacking physiological validation or interpretability analysis.

3. The paper does not sufficiently contrast MnemoDyn against simpler baselines (e.g., recurrent or convolutional models) to justify complexity.

4. Results, though impressive numerically, may be inflated by dataset overlap and lack of external reproducibility tests.

**Questions:**

1. How sensitive is MnemoDyn to the choice of wavelet basis and parameterization depth?

2. Could the operator-based approach generalize effectively to multimodal or task-based fMRI?

3. Does the model yield interpretable components that map to known neural circuits or physiological rhythms?

4. How does MnemoDyn’s performance scale with smaller datasets without fine-tuning?

5. Would the model retain benefits under voxel-level resolution rather than parcellated inputs?

---

> ### Author Response · Authors · 2025-11-21
>
> Dear Reviewer
>
> We thank you for taking the time and effort to carefully review our work and providing valuable feedback. We appreciate the overall positive assessment and seek to clarify the concerns.
>
> **1. Conceptual novelty and relation to prior operator-learning work**
>
> We appreciate the reviewer’s comment. Our goal is not to introduce a fundamentally new mathematical formalism but to adapt these tools in a way that is effective and efficient for modelling rs-fMRI. We clarify the specific architectural contributions that distinguish MnemoDyn from prior operator-learning frameworks—namely, the interaction between wavelet-domain pseudo-differential operators, structured low-rank parameterization, and continuous-time evolution specifically aligned with properties of brain imaging dynamics. Existing operator learning models, like FNO and DeepONet which address a fundamentally different problem, that of supervised operator learning with input-output function pairs data (e.g., PDE forcing -> solution field). They require explicit operator evaluations for training. That is, given the input function $f$, predict the output function $g = G(f)$. MnemoDyn tackled self-supervised sequential modeling of rs-fMRI time-series data; we do not assume access to such paired data. We observe sequences $x(t)$ generated by some unknown dynamical system and must learn representations via a reconstruction objective. Although MnemoDyn contains an operator flavor through its integral formulation, these are categorically different learning problems.
>
> The architectural mismatch follows directly from these problem-formulation differences. FNO parameterizes global Fourier symbols, leading to dense global interactions. This assumption  is well-aligned with PDE solution maps, but less well-suited for multi-scale, nonstationary time series. MnemoDyn, by contrast, operates in a wavelet domain and parameterizes a pseudo-differential operator whose symbol varies across scales. The compact, localized wavelet basis induces approximate banded-diagonal sparsity for long sequences, offering compute and modeling advantages for scale-dependent, nonstationary dynamics that Fourier bases do not naturally provide.
>
> To summarize, canonical operator-learning methods (e.g., FNO) are explicitly designed to learn a map between input–output function pairs, typically arising from PDE solvers. Their architectures, training objectives, and data assumptions all lean on the availability of paired fields, for example, mapping a forcing function to the corresponding PDE solution. MnemoDyn also “learns” an operator but not in a supervised operator map way common in operator learning papers. We operate on long rs-fMRI time series for which no explicit PDE solution, no paired input–output functional data, and no ground-truth operator evaluations are available. The “operator” in MnemoDyn simply arises from the latent integral formulation that parameterizes the evolution rule acting on the hidden state. This is a latent dynamical operator, not a supervised operator-valued regression problem. In this case the kernel acts to modulate the evolution of the latent trajectory, not to map one field to another.
>
> While the underlying principles are drawn from existing theory, their integration yields a foundation model that is markedly different from existing architectures. We have expanded the discussion to articulate this point more clearly.
>
> **2. On neuro-scientific grounding and physiological validation**
>
> We thank the reviewer for bringing up this point. Our approach follows standard latent dynamical-systems abstractions used in rs-fMRI (e.g., state-space and latent ODE models), where BOLD activity is approximated by evolution in a lower dimensional latent space. MnemoDyn extends this by parameterizing the drift via a multi-resolution pseudo-differential operator in the wavelet domain, which naturally decomposes interactions across spatial and temporal scales. This structure yields some interpretability through the learned operator spectrum, even though this does not mean physiological validation. We agree with the reviewer that establishing meaningful neuroscientific grounding is challenging and remains an open problem for almost all data-driven rs-fMRI models, including the baselines. Hence, MnemoDyn cannot be thought of as a neuro-scientific model. We have revised the text to make this clearer in Section 5, and added operator visualizations to ground the discussion in model-level analysis.

---

> > ### Author Response · Authors · 2025-11-21
> >
> > **3. Comparison with simpler baselines**
> >
> > We thank the reviewer for the suggestion. In our experiments, we benchmarked against graph-based and CNN models, which represent the most comparable simple models for whole-brain rs-fMRI sequences. Going further down the tree would require functional-network extraction or highly reduced connectivity features and task-specific network definitions. We have previously explored such reduced representations where one extracts only the networks most associated with the target variable but their performance was consistently below the reported baselines. We are happy to include additional references to this literature from our group and others reporting findings from random forest/decision tree type models on statistical summaries/connectivity measures from different types of resting state networks (e.g., DMN for cognition and DAN for mood/stress).
> >
> > **4. Dataset overlap and external reproducibility**
> >
> > We clarify that pre-training datasets and downstream evaluation datasets are strictly disjoint. There is no overlap. We will release all pre-training code, preprocessing scripts, and model weights to facilitate external reproducibility. This process has already started and we have replicated our findings on two other external datasets (emotion/stress and epilepsy) that were unavailable in pre-training.
> >
> > **5. Sensitivity to wavelet basis and parameterization depth**
> >
> > We thank the reviewer for the question. We had conducted experiments to choose the wavelet basis where we explored different options from the Dabuchies family of wavelet functions. The results indicated very small/negligible variations when exploring the impact of different wavelet families and hence we stick with db2 for all experimental results presented in the paper. Regarding the parameterization depth which depends on the number of scales of wavelet decomposition, we observed that while increasing the number of scales led to faster computation as outlined in Section 2. We choose the number of scales such that the coarsest matrix being materialized had roughly $50 \times 50$ dimensions to give enough expressive power to the model. We mention that the number of scales is dependent on the sequence length of the original input signal. We include a detailed ablation for the choice of wavelet basis functions and the number of levels in the final version of the paper.
> >
> > **6. Generalization to multimodal or task-based fMRI**
> >
> > For multimodal integration (e.g., rs-fMRI + EEG), the framework can in principle handle multi-channel inputs, but this requires non-trivial dataset curation effort, primarily finding cohorts with simultaneous acquisition, which are often much smaller scale.
> >
> > For task fMRI, the challenges are more substantial that we do not know how to handle yet. The biggest problem is heterogeneity. The individual datasets will likely correspond to different tasks (cognitive, emotion etc), different timing protocols (event-related), different stimulus presentation and other confounds. This will likely require either a non-trivial extension or an entirely different approach, and much larger datasets.
> >
> > **7. Interpretability of learned components**
> >
> > We thank the reviewer for this question. The wavelet-based multi-resolution structure gives some interpretability: for example, the learned operator coefficients at different scales correspond to different temporal frequency bands. We visualize which scales dominate the learned dynamics. In the revision, we have added analysis showing the magnitude and sparsity patterns of these coefficients across the decomposition levels. But we avoid claiming correspondence between specific operator components and well-studied scientific concepts (e.g., default mode network, oscillations etc), since such claims would be questionable without strong validation, and beyond our methodological scope. The multi-scale structure offers interpretability only in terms of temporal decomposition; mapping our model (or other transformer or CNN based models) in a deeper way to neuroscience is an open question

---

> > > ### Author Response · Authors · 2025-11-21
> > >
> > > **8. Smaller datasets without fine-tuning**
> > >
> > > We appreciate the suggestion. We have already performed an experiment to demonstrate the effectiveness of MnemoDyn in this regard. To evaluate how **MnemoDyn** scales to smaller datasets without any fine-tuning, we performed two analyses feasible within the time frame of the rebuttal. First, we trained simple downstream MLP heads on the larger HCP-Aging dataset and applied them directly to the dataset collected as part of the Healthy Brain Network (HBN) study. For sex classification, the zero-shot performance reached an accuracy of $0.6393$ and an F1 score of $0.6390$, while for age prediction the MLP achieved an MSE of $0.8516 (0.02)$, indicating moderate transfer for categorical variables but limited transfer for age due to distributional differences. Second, we evaluated the pre-trained MnemoDyn model itself on small datasets by measuring reconstruction quality without any adaptation. A model pretrained on UK Biobank achieved an $R^2$ of $0.98$ on HBN and $0.96$ on HCP-Aging, with corresponding MSEs of $7.89 \times 10^{-8}$ and $4.52 \times 10^{-8}$, respectively. These results show that MnemoDyn’s learned temporal operators generalize extremely well to small datasets, maintaining high reconstruction accuracy even in the absence of fine-tuning. These results illustrate that the architectural design allows the model to retain and transfer a substantial portion of its representational benefit. We will include this in the final version of the paper once the discussion phase draws to a close.
> > >
> > > **9. Extension to voxel-level resolution**
> > >
> > > Even at the lower end, voxel-level rs-fMRI will have $200K$ voxels per volume, plus time. Most SOTA video foundation models operate on heavily downsampled inputs due to compute constraints, and video datasets are many orders of magnitude larger. We suspect that with our dataset scale, voxel-level modeling would face overfitting unless we have larger datasets.
> > >
> > >
> > > If the clarification(s) above addresses the reviewer’s concern, we will upload the updated manuscript promptly to reflect these revisions.

---

> > > > ### Author Response · Authors · 2025-12-04
> > > >
> > > > We thank the reviewer for their time, thoughtful feedback and constructive suggestion(s) which have improved the presentation of our work. We have uploaded a revised pdf with the suggested and promised changes, marked in blue.

---

### Official Review · Reviewer_Yupk · 2025-10-31

**Soundness:** 2
**Presentation:** 3
**Contribution:** 2
**Rating:** 2
**Confidence:** 4

**Summary:**

The paper introduces a continuous-time latent dynamical model that learns an evolution operator over rs-fMRI trajectories, parameterized via multi-resolution wavelet kernels and implemented with pseudo-differential operators to induce sparsity and block-diagonal computation.​

Pretrained on ~40K rs-fMRI sequences and fine-tuned with lightweight heads, MnemoDyn improves reconstruction, clinical classification (ADNI), and demographic/trait prediction (UKB, HCP-Aging) versus other approaches, such as Brain-JEPA, BrainLM, and graph CNNs, while requiring less computing power.​

**Strengths:**

- The paper proposes an alternative to the tokenization/positional heuristics problems, which are common to transformer-based methods.

- Efficient multiscale design: wavelet-domain pseudo-differential parameterization and CP tensor factorization exploit sparsity and low-rank structure, enabling single-GPU pretraining and potentially allowing scaling to longer sequences than conventional transformer-based methods can handle.​

- Broad and consistent gains: large improvements on ADNI NC/MCI and amyloid classification, and better age/sex/trait prediction on HCP-Aging and other cohorts (HBN, ADHD-200, NKIR), indicating robust transfer across sites/protocols.​

**Weaknesses:**

- Operator interpretability is under-explored: the paper motivates pseudo-differential structure and multiscale kernels but provides limited analysis of learned symbols/kernels or neurophysiological correlates of the operator.​

- Pretraining objectives comparison is shallow: masking and JEPA-style training are included, but ablations isolating what aspects of the operator parameterization versus objective drive gains are not fully disentangled.​

- The figures in the paper do not help with clarification regarding model structure, training and inference.

**Questions:**

- Figure 3 presents parcel reconstructions for unseen data, where the predicted signals show an almost perfect overlap with the measured fMRI time series. Since this is unseen data (ie, not seen during training), this raises concerns about how inference is performed with this model. Are these outputs generated by evolving the latent state from an initial condition (ie, first point of the dynamics)? Is this the result of solving equation 11? If the reconstruction indeed follows the ODE solution over time, it is surprising that the model can reproduce signals with noise-like structure, such as those shown for the UK Biobank example in Fig 3. More details on the inference procedure are necessary to clarify this result.

- The authors motivate their model using a continuous-time ODE formulation. However, modeling brain dynamics as a first-order (Markovian) system may be restrictive, given the nonlocal and temporally dependent nature of neuronal interactions. Could the authors elaborate on why ODEs, which assume locality in time, were chosen over formulations that can capture non-Markovian dependencies, such as integral equation (IE) or delay-based models? Prior work has shown IE formulations can yield superior numerical accuracy even for Markovian dynamics [1, 2, 3]; therefore, additional justification for the ODE assumption would be valuable.

- Since MnemoDyn is conceptually related to Fourier Neural Operator (FNO) and DeepONet—both of which learn functional mappings using spectral or integral representations—it would strengthen the work to include direct comparisons with these models. In particular, given that MnemoDyn operates in the wavelet domain (analogous to Fourier parameterization in FNO), a head-to-head evaluation would help quantify the benefits of the proposed multiresolution pseudo-differential operator design.

- Tables 2 and 3 report MnemoDyn outperforming several transformer and graph-based baselines on clinical and demographic prediction tasks. Were all baseline models fine-tuned under the same conditions and on the same datasets as MnemoDyn? Additional details about training protocols, data splits, and hyperparameter parity are necessary to assess the fairness of these comparisons.
The paper refers to MnemoDyn as a “foundation model.” Could the authors clarify the specific criteria under which this designation applies? For instance:
  - Does MnemoDyn show improved performance with scale (in model or data size)?
  - Can its representations transfer effectively to other neuroimaging modalities or tasks beyond label prediction?
  - How does it satisfy the general-purpose, reusable, and cross-domain characteristics typically associated with foundation models?

Ref:

[1] Vladimir Rokhlin. Rapid solution of integral equations of classical potential theory. Journal of computational physics, 60(2):187–207, 1985.

[2] Vladimir Rokhlin. Rapid solution of integral equations of scattering theory in two dimensions. Journal of Computational Physics, 86(2):414–439, 1990.

[3] Zappala, Emanuele, Antonio Henrique de Oliveira Fonseca, Josue Ortega Caro, Andrew Henry Moberly, Michael James Higley, Jessica Cardin, and David van Dijk. "Learning integral operators via neural integral equations." Nature Machine Intelligence 6, no. 9 (2024): 1046-1062.

---

> ### Author Response · Authors · 2025-11-21
>
> Dear Reviewer,
>
> We appreciate the time spent in reviewing our work and providing details of what you believe are deficiencies. Below, we clarify all your questions.
>
> **1.Interpretability of learned pseudo-differential operator**
>
> Thank you for this suggestion. Based on this comment and suggestion from reviewer oX9c, we now include an analysis examining the decomposition of the operator A into its diagonally banded components and the small dense term that operates across scales. We report parameter norms across different wavelet levels to highlight which multiresolution components become dominant after pretraining. In addition, we visualize representative activations arising from the interaction between A and the input signal at each scale, showing how different temporal bands influence the learned dynamics. These analyses show at a minimum that the wavelet-domain parameterization provides structured decomposition of the learned dynamics across temporal scales.
>
> We should emphasize a distinction: while the multi-resolution structure offers more direct interpretability than black-box attention mechanisms, we avoided claiming some sort of correspondence between operator components and specific neural circuits, as such claims would need careful validation beyond the scope of this modeling-focused paper. The wavelet decomposition does give interpretability in terms of temporal structure - which scales dominate the dynamics and how the operator distributes influence across time. But this is not necessarily in terms of specific brain networks. The reviewer will likely agree that this is already more interpretable than the attention patterns in Brain-JEPA or BrainLM, neither of which include such analyses in their papers. That said, we agree that a deeper investigation of neurophysiological correlates is valuable future work once the architecture/model, our main contribution, is established and adopted by the community.
>
> **2.Comparison of Pre-training Objective**
>
> To clarify this doubt, we note that the existing results already provide strong evidence that the operator-based architecture drives the gains, and this is not coming from the specific pre-training objective. We briefly explain why. Tables 2--3 show that both MnemoDyn-Mask and MnemoDyn-Mask-JEPA achieve similar performance across diverse tasks. Consider for example, on HCP-Aging age prediction both achieve MSE$=0.90$, and on ADNI NC/MCI classification they obtain ~$96$% vs ~$94$% accuracy. This is consistent across different objectives and shows that our formulation is the primary contributor. More importantly, both variants outperform the baseline methods that inspired each objective: MnemoDyn-Mask exceeds BrainLM (which uses masking) by ~$4$% on HCP-Aging sex classification, while MnemoDyn-Mask-JEPA exceeds Brain-JEPA by ~$16$% on ADNI NC/MCI classification task. These margins with matched objectives isolate the architectural contribution.
>
> To further support this finding, we are happy to add an ablation using a simple denoising autoencoding as a third pre-training variant in the revision, which we will upload once your concerns are fully resolved. We expect this will similarly achieve strong performance, reinforcing that the formulation not careful engineering underlies the reported improvements. We believe that the behavior _across_ objectives is actually a signal that the operator formulation is giving the right inductive biases regardless of the surface-level training objective.
>
> **3. Improved and more detailed Figure(s)**
>
> We agree that more detailed visualization would strengthen the paper. In the revision, we will add Figure 2 (new) to the main text summarizing the detailed architecture with three panels. Panel A will show the Multi-resolution operator structure with a diagram of the wavelet decomposition across scales. Basically, showing how input signals are decomposed into temporal bands, transformed by scale-specific operators, and recombined. This will clarify the multi-resolution temporal modeling which is mentioned but not shown visually. Panel B will show the training inference flow. This will show the side-by-side comparison showing pre-training with spatiotemporal masking, then input with $70$% masked region, then encoder followed by operator evolution and then decoder followed by reconstruction loss on masked regions only. On the right, we will show fine-tuning. This should address the confusion about inference. Suggestions welcome. Finally, Panel C will show the block-diagonal computation, visualization of the CP tensor factorization and resulting sparse operator structure.
>
> We are happy to enhance Figure 1 by replacing the schematic with a more detailed end-to-end pipeline that shows the actual data flow. This keeps the overview accessible while adding details that were described in text.

---

> > ### Author Response · Authors · 2025-11-21
> >
> > **4.Visualization of Reconstruction Output**
> >
> > Thank you for asking us to clarify this. We emphasize that (11) operates in latent space, not directly in observation space. The encoder and decoder give us the bridge between these spaces, which is why the reconstruction is not a direct ODE solution in observation coordinates. We will further clarify this point in the revision. The inference procedure is as follows: The parcellated rs-fMRI sequence with say $70$% spatiotemporal masking is fed to the encoder. This then projects the masked input to the latent state. The evolution operator produces the latent trajectory. The decoder maps the latent trajectory back to the observation space where the loss is calculated only on the masked regions. The model never sees these regions during the forward pass and must estimate them from the spatiotemporal context. The ability to reconstruct these components from 30% context shows that the learned operator captures relevant structure in what appears locally as noise. In order to address the reviewer concern, regarding reconstruction quality we mention that the figure is from an auto-encoding setup and not the masked auto-encoding pre-training procedure which is what we use for experiments in the paper. We have now updated the figure with the figure from the masked pre-training strategy to avoid any confusion.
> >
> > **5. Why an ODE formulation for rs-fMRI dynamics and not a non-Markovian integral- or delay-based model?**
> >
> > This is an important question. We clarify that while we motivate the model using ODE notation for simplicity, MnemoDyn implements non-Markovian dynamics through its integral operator formulation. The reviewer’s concern about first-order Markovian systems would be valid if we directly worked with Eq. (3), but this is not what we do.
> > Indeed, we motivate the setup using the following ODE (Eqn 3):
> > $$
> > \frac{dz(t)}{dt} = F(z(t), u(t); \theta)
> > $$
> > But if you check Equations 7–16, you will immediately see that the dynamics are governed by the integral operator and are not first-order Markov as described below.
> > Integrating Eqn (3) yields the operator form in Eqn (7):
> > $$
> > z(t) = z_0 + \int_0^t F(z(\tau), u(\tau); \theta) d\tau
> > $$
> > Which is decomposed and written in Eqn (11) as:
> > $$
> > z(t) = z_0 + \int_0^t P(z(\tau); \theta) d\tau + (K_\theta u)(t)
> > $$
> > This integral from 0 to t is important: at each time point, the operator $K_\theta$ has access to the entire history of the input trajectory $u(\cdot)$ from the initial time to the current time. After this point, we represent the kernel of the operator $K$ using separable wavelet basis which leads to Eqn (15):
> > $$
> > (K_\theta u)(t) = \sum_{j,k} \int_0^t A_{j,k}(z(\tau)) u_{j,k}(\tau) d\tau
> > $$
> >
> > There is an explicit integration of the signal projected on the wavelet basis. The wavelet decomposition provides multi-scale access to this history where coarse scales integrate over long temporal windows and fine scales focus on recent local structure. This is explicitly non-Markovian: the instantaneous rate $dz/dt$ depends on the integral over the past trajectory, not just the current state $z(t)$. Thus, even though the evolution is written in ODE form, the model evaluates $F$ on wavelet-integrated summaries of the entire past trajectory. Finally, the operator couples all these nonlocal integrals across space (different RoIs) and time together to modulate the latent dynamics of rs-fMRI.
> >
> > Additionally, to make the analysis even more clearer, one can re-write Eqn(11) as:
> > $$
> > z(t) = z_0 + \int_0^t P(z(\tau)), d\tau + \int_0^t K(z(\tau)), d u^{W}(\tau)
> > $$
> > Where $u^{W}(\tau)$ is the multi-scale control path associated with the corresponding Controlled Differential Equation (CDE). In CDE terminology, the wavelet-transformed path serves as the so-called “rough path” that encodes history beyond point-wise evaluation. The reviewer will likely recognize that CDEs are a generalization of ODE, precisely to capture non-Markovian dependencies, which is what MnemoDyn implements.
> >
> > We hope that the reviewer now appreciates the precise modelling choices made in **MnemoDyn** and why it is a good fit. We appreciate the references which we will add. Regarding the connection to integral equations: we emphasize that our formulation in (9)-(11) is an integral equation with a learned kernel $K$. The actual model also implements the integral operator form. The main design choice is how we parameterize this integral kernel. We appreciate the references to integral equation methods, which are also focused on efficiency but with known kernels. The reference to Zappala.et.al on neural integral equations is more relevant, but addresses supervised operator learning with paired input-outputs. These are somewhat different problem settings but working out deeper links between them is nonetheless interesting.

---

> > > ### Author Response · Authors · 2025-11-21
> > >
> > > **6.Similarity and Distinctions Operator Learning**
> > >
> > > We want to take this opportunity to clarify a very important distinction, which appears to be a major score driver. FNO and DeepONet address a fundamentally different problem, that of supervised operator learning with input-output function pairs data (e.g., PDE forcing -> solution field). They require explicit operator evaluations for training. That is, given the input function $f$, predict the output function $g = G(f)$. MnemoDyn tackled self-supervised sequential modeling of rs-fMRI time-series data; we do not assume access to such paired data. We observe sequences $x(t)$ generated by some unknown dynamical system and must learn representations via a reconstruction objective. Although MnemoDyn contains an operator flavor through its integral formulation, these are categorically different learning problems.
> > >
> > > The architectural mismatch follows directly from these problem-formulation differences. FNO parameterizes global Fourier symbols, leading to dense global interactions. This assumption  is well-aligned with PDE solution maps, but less well-suited for multi-scale, nonstationary time series. MnemoDyn, by contrast, operates in a wavelet domain and parameterizes a pseudo-differential operator whose symbol varies across scales. The compact, localized wavelet basis induces approximate banded-diagonal sparsity for long sequences, offering compute and modeling advantages for scale-dependent, nonstationary dynamics that Fourier bases do not naturally provide. DeepONet uses two different sub networks without providing an explicit interpretation of the projection basis thereby making it harder to justify its choice for modelling multi-scale spatio-temporal rs-fMRI data.
> > >
> > > To summarize, canonical operator-learning methods (e.g., FNO) are explicitly designed to learn a map between input–output function pairs, typically arising from PDE solvers. Their architectures, training objectives, and data assumptions all lean on the availability of paired fields, for example, mapping a forcing function to the corresponding PDE solution. MnemoDyn also “learns” an operator but not in a supervised operator map way common in operator learning papers. We operate on long rs-fMRI time series for which no explicit PDE solution, no paired input–output functional data, and no ground-truth operator evaluations are available. The “operator” in MnemoDyn simply arises from the latent integral formulation that parameterizes the evolution rule acting on the hidden state. This is a latent dynamical operator, not a supervised operator-valued regression problem. In this case the kernel acts to modulate the evolution of the latent trajectory, not to map one field to another.
> > >
> > > Finally, the reviewer will agree that designing a  FNO or DeepONet based foundation model for rs-fMRI would require: (a) new objectives for self-supervised sequence modeling, (b) basis replacement or some hybrid choice, (c) adding structure to the operator to account for brain parcellations and irregular sampling, and (d) neuroimaging-specific regularization and masking. Such modifications will amount to a full redesign of a  FNO/DeepONet inspired model for our problem, rather than a direct comparison with an existing method. Therefore, we prioritized comparisons to **strong, directly applicable, domain-specific rs-fMRI** foundation model baselines including recent models published at major venues in our community.
> > >
> > > **7.Comparison with baseline**
> > >
> > > Yes, all baselines were fine-tuned under identical conditions, including dataset split to ensure fair comparison. All these details for reproducibility will be available on the GitHub repository where we will release the code/model weights.

---

> > > > ### Author Response · Authors · 2025-11-21
> > > >
> > > > **8.Why MnemoDyn is a Foundation Model?**
> > > >
> > > > (a) Does **MnemoDyn** show improved performance with scale (in model or data size)?
> > > >
> > > > Yes, in both model and data dimensions. Indeed MnemoDyn shows improved performance with increased model size. As mentioned in the implementation details, we observed a clear improvement in performance when stacking blocks of the proposed model in a similar fashion as stacking transformer blocks in LLMs (four operator blocks versus two). For increasing dataset size, we observe a similar behavior where pre-training with 80% of UK-Biobank was more effective than pre-training with half of the dataset size.
> > > >
> > > > (b) Can its representations transfer effectively to other neuro-imaging modalities or tasks beyond label prediction?
> > > >
> > > > Yes, we should clarify the scope. Regarding tasks, label prediction (classification and regression) is the established evaluation protocol for rs-fMRI foundation models, used consistently by Brain-JEPA and Brain-LM. This allows direct, fair comparison with strong baselines in our community on scientifically relevant tasks. But MnemoDyn is set up as a general-purpose representation learner. The latent trajectories z it produces summarize multi-scale temporal dependencies and spatial interactions, and can be easily used with lightweight task-specific heads for a wide range of downstream objectives (e.g., regression, trajectory forecasting, or contrastive and clustering-style objectives) without modifying the backbone.
> > > > Regarding transfer to other modalities, we clarify what is reasonable/feasible within a single paper.  The reviewer may be underestimating the scope of work already presented. Rs-fMRI is one of the most widely acquired neuro-imaging modalities in large-scale studies making a pre-trained model for this modality directly impactful for a large scientific community. More importantly, our current experiments span both public and restricted-access datasets, each needs separate institutional approvals, data use agreements, and compliance. These processes take up to months per dataset. Even after approval, data transfer restrictions mean that experiments must run on limited compute resources provided by the data custodian, not on our infrastructure. This process of dataset negotiation, access approval, preprocessing, and pre-training for the results presented here required 6+ months of institutional coordination and experiment/compute work. Extending to another modality (EEG, MEG) would require a comparable 6-9 month effort simply for dataset curation. This represents a full follow-up project, not an additional experiment.
> > > >
> > > > (c) How does it satisfy the general-purpose, reusable, and cross-domain characteristics typically associated with foundation models?
> > > >
> > > > We thank the reviewer for the question. Within the scope of rs-fMRI, a popular modality in large-scale neuro-imaging studies, MnemoDyn satisfies the core features of a foundation model: general-purpose, reusable, and cross-domain within the modality, where by cross-domain we mean variation in scanners, sites, demographics, and health conditions. As shown in Tables 1 and 2, its pre-trained representations transfer effectively across multiple downstream regression and classification tasks, showing that the features it learns are both general-purpose and easily reusable with lightweight task-specific heads. Moreover, MnemoDyn performs consistently across heterogeneous datasets, indicating strong cross-domain robustness within rs-fMRI.
> > > >
> > > > Overall, the observed scaling behavior, successful cross-dataset and cross-modality transfer, and the model’s strong performance across diverse downstream tasks demonstrate that MnemoDyn exhibits several core properties of a foundation model for large-scale neuro-imaging, a term referred to in both recent baseline papers of BrainLM and Brain-JEPA.
> > > >
> > > > If the clarification(s) above addresses the reviewer’s concern, we will upload the updated manuscript promptly to reflect these revisions.

---

> > > > > ### Comment · Reviewer_Yupk · 2025-11-28
> > > > >
> > > > > - Q1 (Interpretability):
> > > > >
> > > > > The planned decomposition of A into banded and dense cross‑scale components, the reporting of parameter norms across wavelet levels, and the scale‑wise activation visualizations directly address my request for basic interpretability of the learned pseudo‑differential operator. I also appreciate the clear distinction that the authors draw between temporal/multiscale interpretability and stronger neurophysiological claims. As long as these analyses and caveats appear in the text, I consider my concern about operator interpretability satisfactorily addressed.
> > > > >
> > > > > - Q2 (Comparison of pre-training objectives):
> > > > >
> > > > > I agree with the authors' comments on the MnemoDyn‑Mask and MnemoDyn‑Mask‑JEPA outperforming their objective-matched baselines as a suggestion that the architecture contributes to the gains. However, similar performance across two objectives indicates robustness but does not fully disentangle the role of the operator parameterization from other factors such as optimization details, capacity, or head design. Likewise, improvements over BrainLM/Brain‑JEPA assume comparable tuning and effective capacity, which is not fully documented in the current version.
> > > > > I therefore welcome the authors’ plan to add a third variant based on a simple denoising autoencoding objective. I would encourage them to (i) include this ablation in the revised manuscript and (ii) explicitly present the conclusion as “suggestive but not definitive”: i.e., that across several substantially different pre‑training objectives, the architecture appears to dominate performance, while acknowledging that a complete separation of objective vs. architecture effects is beyond the current experimental scope. A brief discussion along these lines would help readers correctly interpret the strengths and limitations of the pre‑training objective comparison.
> > > > >
> > > > > - Q3 (Improved figures):
> > > > >
> > > > > I appreciate the planned changes to the figures and consider this point addressed, conditional on the revised figures making the architecture and training/inference flows as clear as described.
> > > > >
> > > > > - Q4 (Reconstruction and inference):
> > > > >
> > > > > Thank you for the clarification regarding Figure 3. This explanation resolves my main concern about potential leakage in the forward pass. However, in the current version, the text and original figure make this very difficult for a reader to infer. I strongly encourage the authors to (i) clearly separate the “plain autoencoding” setup from the masked‑autoencoding pre‑training used in the main experiments, and (ii) add a concise description of the inference pipeline in the main paper (encoder → latent evolution operator → decoder, with loss only on masked regions), not only in the appendix. Updating Figure 3 to reflect the masked setting, as you propose, will also make the message much clearer.
> > > > >
> > > > > - Q5 (ODE vs non‑Markovian / integral formulation)
> > > > >
> > > > > Your response clarifies that the implemented model is not a simple first‑order Markov ODE, but an integral‑ or CDE‑style evolution where the latent dynamics depend on a history‑encoding control path in the wavelet domain. This addresses my original concern about a strictly Markovian assumption. However, this distinction is not currently evident in the main text: the paper primarily motivates the method as an ODE model of brain dynamics, and only briefly hints at the integral‑operator view.
> > > > > To make this clear for readers, I would recommend two concrete changes:
> > > > >
> > > > > 1) Explicitly state in the main modeling section (not only around Eqs. 7–16) that the effective evolution depends on an integral over the past trajectory and is therefore non‑Markovian. If possible, include the controlled differential equation interpretation already mentioned in your response (wavelet‑transformed path as control/rough path), which provides a clean conceptual handle.
> > > > >
> > > > > 2) Add a short discussion of the modeling choice relative to explicit IE/NIE‑style approaches. You rightly note that your formulation is an integral equation with a learned kernel, and that the key design choice is how this kernel is parameterized. A brief explanation of why you chose this ODE/CDE + wavelet parameterization over alternative integral formulations (e.g., neural integral equations with explicit IE solvers) at the conceptual level (numerical behavior, long‑sequence suitability, computational considerations) would help situate MnemoDyn for readers familiar with that literature.

---

> > > > > > ### Comment · Reviewer_Yupk · 2025-11-28
> > > > > >
> > > > > > - Q6 (Relation to FNO / DeepONet and operator learning):
> > > > > >
> > > > > > I appreciate the distinction you draw between classical supervised operator learning (FNO, DeepONet) and your self‑supervised sequence modeling setting. It is fair that off‑the‑shelf FNO/DeepONet, designed for PDE input–output maps, are not directly applicable to rs‑fMRI pretraining. At the same time, the underlying architectural ideas (spectral kernel parameterization, nonlocal integral operators) can, in principle, be adapted to self‑supervised dynamics learning by changing the objective and how training pairs are constructed. In this light, describing these methods as “categorically different” feels stronger than necessary.
> > > > > > I still see value in positioning MnemoDyn explicitly as one particular instantiation within the broader neural operator / CDE / SSM design space, and, where feasible, providing evidence that your specific choices (wavelet pseudo‑differential parameterization, multi‑scale symbol structure) matter relative to natural alternatives. This need not be a full FNO/DeepONet benchmark on rs‑fMRI; even controlled ablations—such as a Fourier‑basis variant within your framework, or a simpler CDE/SSM‑style kernel—would significantly strengthen the architectural claims and help disentangle gains due to the operator parameterization from those due to a well‑trained CDE/SSM‑like backbone.
> > > > > > Finally, given your response to Q5, where you frame MnemoDyn as implementing an integral operator with a learned kernel and connect it to CDEs, it would be helpful to briefly situate your method relative to integral‑equation frameworks such as Neural Integral Equations. While I do not expect a full NIE benchmark on rs‑fMRI, a short comparison of (i) problem setup (supervised vs self‑supervised), (ii) solver choice (ODE/CDE vs IE), and (iii) kernel parameterization, together with a brief justification for preferring your wavelet pseudo‑differential construction over IE/NIE‑style formulations, would make the claims about architectural novelty and operator learning more compelling.
> > > > > >
> > > > > > - Q7 (Comparison with baselines):
> > > > > >
> > > > > > Thank you for confirming that all baselines were fine‑tuned under identical conditions and dataset splits. To assess fairness and reproducibility from the paper alone, it would be very helpful to add a more concrete description of the baseline protocols: for example, the head architectures used for each model, the hyperparameter search ranges and budgets, and how pretraining differences (for foundation‑style baselines) were handled. Without at least a brief summary of these choices in the main text or appendix, it remains difficult for readers to evaluate how much of the reported gains may be attributable to the MnemoDyn architecture versus differences in tuning effort or training setup.

---

> > > > > > > ### Comment · Reviewer_Yupk · 2025-11-28
> > > > > > >
> > > > > > > - Q8 (Why MnemoDyn is a “foundation model”):
> > > > > > >
> > > > > > > Thank you for the additional clarification.
> > > > > > >
> > > > > > > 1) I appreciate the statement that MnemoDyn improves with both model depth (more operator blocks) and pretraining data (80% vs 50% of UK Biobank). However, this is not yet visible in the current paper. If scaling behaviour is part of the argument for calling MnemoDyn a foundation model, it would be important to include at least a brief quantitative summary (e.g., a small table or figure with 2–3 model sizes and 2–3 data fractions) so readers can see the trend rather than relying on anecdotal statements.
> > > > > > > 2)  Regarding transfer, all experiments shown are supervised label‑prediction tasks within a single modality (rs‑fMRI). I understand and respect the practical constraints around adding EEG/MEG experiments, and you already note that such extensions would be a separate project. Empirically, the current evidence therefore supports cross‑dataset transfer and multi‑task reuse within rs‑fMRI, rather than cross‑modality transfer or non‑label objectives. It would strengthen the claims to either (i) add at least one non‑label downstream objective (e.g., forecasting or clustering) within rs‑fMRI, or (ii) explicitly narrow the wording to a “foundation model for rs‑fMRI” and clearly flag any cross‑modality aspects as future work.
> > > > > > > 3) Your operational definition of a foundation model is effectively within‑modality: a large, self‑supervised backbone that (i) is pretrained at scale, (ii) supports multiple downstream tasks with lightweight heads, and (iii) is robust across datasets, scanners, and populations in that modality. Within that scope, the results do show promising “general‑purpose and reusable” behaviour across multiple rs‑fMRI cohorts and tasks. At the same time, this is narrower than how the term is often used in the broader ML literature, where foundation models typically exhibit broader task coverage and, in many cases, multi‑modality.
> > > > > > >
> > > > > > > To avoid overclaiming, I would encourage you to make this scope explicit in the paper: clearly state that (1) the demonstrated generality is across rs‑fMRI datasets and label‑based tasks, and (2) cross‑modality properties are, at this stage, only hypothesized. Since scaling behaviour is central to your argument, including a brief quantitative summary of the model‑size and data‑size scaling results (e.g., 2 vs 4 operator blocks and 50% vs 80% UKB) would also be important. A short paragraph with this calibrated framing, together with concrete scaling evidence, would help readers correctly interpret both the strength and the limits of the model designation in this work.

---

> ### Author Response · Authors · 2025-12-04
>
> We sincerely thank the reviewer for their time, thoughtful feedback, and constructive suggestions. We are glad that our responses have clarified the concerns. In the revised document we have fully incorporated the suggested and promised changes.
>
> We have uploaded a revised manuscript, with all updates marked in blue. Below, we summarize the key modifications made:
>
> 1. **Interpretability of the learned operator.** We have included a section on analysis of the learned operator in a paragraph named “Discussion” (page 9) of the main paper and additional detail(s) and visualization(s) have been added in Appendix A.3.
>
> 2. **Neurophysiological claims.** We have clarified the scope of MnemoDyn in this regard in Remark 2.4 (page 6) and in the Conclusion section in the main paper.
>
> 3. **Runtime and memory comparison.** We have revised the content in Remark 3.1 (page 7)
>
> 4. **$R^2$ score for Reconstruction performance.** We have included the (R^2) reconstruction scores in Table 1 (page 7).
>
> 5. **Sensitivity to wavelet basis.** We have clarified the sensitivity to the choice of wavelet basis in Implementation Details (page 6).
>
> 6. **Baseline clarity.** We have further clarified our choice of baseline(s) in Section 3.4 (page 9)
>
> 7. **Transfer without fine-tuning.** We have added a new experiment to check the zero shot performance of MnemoDyn in Appendix A.8 (page 25) and the results are promising
>
> 8. **Improved visualizations.** We have added an entirely new Figure 4 to demonstrate the pre-training and fine-tuning training paradigms.
>
> 9. **Masked Pre-training.** We have highlighted the masked pre-training strategy on page 8 and in Appendix A.2.3 (page 18).
>
> 10. **Updated Figure 3.** We have updated Figure 3 to show reconstruction results from masked pre-training output apart from clarifying the objective as mentioned before.
>
> 11. **Ablations.** Ablations for removing multi-scale structure and discussion on low-rank components are now included in Appendix A.6 (page 23).
>
> 12. **Positioning as a foundation model.** We have clarified the scope of MnemoDyn as an rs-fMRI foundation model, added numerical results for scaling with model size (Table 5) and dataset size (Table 6) on page 21. We have discussed what “general-purpose” means in this domain, and limits of cross-modal generalization in Remark 2.5 (page 6) and Appendix A.4 (page 20).
>
> 13. **Relation to operator learning.** We now have a strengthened discussion distinguishing MnemoDyn’s self-supervised sequential learning from supervised operator-learning frameworks briefly in Remark 2.3 (page 4) and more elaborately in Appendix A.5 (page 22).
>
> 14. **Connection to CDE/IE.** We have demonstrated clearly how MnemoDyn implements a CDE (which is a generalization over ODE), has non-Markovian characteristics and discussed its motivation going beyond IE formulations throughout Section 2.
>
> 15. **Hyper-parameter Tuning.** We have added details of hyperparameter search in Appendix A.2.1 (page 17).
>
> 16. **Denoising auto-encoding pre-training.** We have mentioned the use of denoising auto-encoding as a viable pre-training strategy and added new results for the same in Table 3 (page 8).
>
> We hope these revisions clearly demonstrate that we have addressed every concern raised and strengthened the manuscript in both clarity and technical depth. We sincerely appreciate the reviewers’ input, which have meaningfully improved the quality and presentation of our work.

---

### Official Review · Reviewer_oX9c · 2025-10-31

**Soundness:** 4
**Presentation:** 3
**Contribution:** 3
**Rating:** 6
**Confidence:** 4

**Summary:**

This paper introduces MnemoDyn, a multiscale state-space model designed to model resting-state fMRI data. The model uses a wavelet basis to represent temporal dynamics and a structured, learnable transition matrix A to capture latent state evolution. The approach keeps the multidimensional structure of fMRI data and shows notable improvements in fine-tuning tasks compared to existing baselines.

**Strengths:**

•	Creative use of multiscale state-space modeling for fMRI signals.
•	Preserves spatial correlations across brain regions rather than treating them independently.
•	Demonstrates consistent fine-tuning performance gains over strong baselines.
•	The model seems more parameter-efficient than transformers, which is valuable for neuroimaging data

**Weaknesses:**

•	The claimed computational efficiency isn’t empirically supported—there are no runtime or scalability benchmarks.
•	Missing interpretability analysis of the learned transition matrix A (e.g., identifying key frequencies or scales).
•	Table 1 lacks R² reconstruction metrics for HCP and UK Biobank datasets, which would be a necessary metric for comparisons.
•	The trade-off between pretraining effort and downstream performance is mentioned but not shown.
•	Writing could be tightened in the methods section for clarity.

**Questions:**

1.	Can you include runtime and memory benchmarks to support the efficiency claims?
2.	Could you analyze the learned matrix A to show which frequencies or scales are most predictive?
3.	Please report R² reconstruction results for HCP and UK Biobank, ideally alongside fine-tuning scores.
4.	Could you visualize or quantify the trade-off between pretraining difficulty and fine-tuning accuracy?
5.	How sensitive is the model to the choice of wavelet basis or number of scales?


Suggestions:
•	The methods section could benefit from a short schematic showing how the wavelet decomposition and latent transition components interact.
•	Consider adding an ablation study (e.g., removing multiscale structure or low-rank decomposition).

---

> ### Author Response · Authors · 2025-11-21
>
> Dear Reviewer
>
> We thank you for taking the time and effort to review our work and providing valuable feedback. We appreciate the overall positive assessment and seek to clarify the concerns.
>
> **1. Runtime and Memory benchmarks**
>
> Thanks for the question. The efficiency gap between **Mnemodyn** and our baselines is rather significant. On a single 40 GB GPU, pre-training for MnemoDyn on UK-Biobank converges in 100 epochs taking ~5.5 hours total. This model has 92M parameters and peak memory usage is ~20GB. Our baseline papers do not report time/memory requirements in the main paper or supplement. The Brain-JEPA paper does note that they use 4 A100 GPUs, so our pre-training needs of a single 40 GB GPU are already advantageous. For pre-training Brain-JEPA on UK-Biobank with the ViT-base model on 4 A100 GPUs (40GB each) the total training time is ~16.5 hours, which is again 4 times more than MnemoDyn despite utilizing a much larger compute. BrainLM resource needs are at least as much as Brain-JEPA. But if suggested, in the final version, we will include a dedicated subsection reporting runtime, memory consumption, relative to all these transformer baselines under identical training setups.
>
> **2. Analysis of learned matrix (A)**
>
> Thank you for the suggestion! To be responsive to this question, we ran experiments where we decomposed A into several diagonally banded matrices and a small but dense matrix across different level(s). We use a parameter norm for different levels of decomposition to show which ones are more dominant than the others after the pre-training phase. We summarize the main observations here.
>
> The Frobenius-norm analysis shows a clear and highly consistent multi-scale structure that directly reflects the operator parameterization in MnemoDyn. We see strong concentration of norm mass in the kernels from the wavelet-domain pseudo-differential operator. This suggests that the model’s dynamics are governed mostly by these structured, cross-scale filters and not unstructured dense mappings. In contrast, the dense layers have much smaller norms that decrease with depth, indicating that deeper blocks rely less on local, single-scale corrections and increasingly on global, long-range structure captured by the wavelet operator. The components that directly handle the raw input steadily decrease in magnitude with depth. Overall, these trends provide good evidence that MnemoDyn parameterization explicitly induces and preserves the multi-scale structure of the underlying rs-fMRI dynamics.
>
> The sparsity patterns across reinforce the same multi-scale picture revealed by the norm analysis. The kernels—the components responsible for implementing the pseudo-differential operator—are consistently and uniformly dense across all layers. This is exactly what we expect from a learned operator that must capture smooth, global, cross-scale transformations: such operators do not benefit from element-wise pruning, and the model indeed preserves them in fully dense form. By contrast, the only component exhibiting extremely high sparsity is the output-projection matrix, which becomes over 95% sparse. This indicates that once the multi-scale transformation has been applied, only a very small subset of coordinates is needed to map the internal multi-resolution representation back to the lower-dimensional space. In other words, the learned operator is rich and distributed, but the readout from this operator is highly selective—consistent with the idea that the dynamics live on a structured, lower-dimensional manifold extracted through the wavelet-based parameterization.
>
> We have added this discussion and believe that these additional details in the updated manuscript will be helpful.
>
> **3. Missing $R^2$ reconstruction metrics**
>
> This is another great suggestion as it helps us talk about reconstruction quality. In the updated draft, we now report both MSE and $R^2$ for UK Biobank and HCP, including cross-dataset evaluations in Table 1. Mnemodyn-UKB reaches an $R^2$ of $0.98475$ on UKB and $0.93425$ on HCP, whereas Mnemodyn-HCP achieves $0.986723$ on HCP and $0.96872$ on UK-Biobank demonstrating strong in-dataset performance with good cross-dataset transfer. We appreciate the idea.

---

> > ### Author Response · Authors · 2025-11-21
> >
> > **4. Trade-off between pre-training cost and downstream performance**
> >
> > To address the reviewer’s question about the role of pre-training effort, we conducted an additional experiment that traces downstream task performance as a function of pre-training duration. We summarize the main findings here. Across checkpoints from different stages of pre-training, we observe a clear and consistent improvement in downstream performance as pre-training progresses. In the time frame of this author feedback phase, we focused on sex-classification for the HCP-Aging dataset. At $5$ epochs, the model performs modestly (Acc $= 0.61$, F1 $= 0.60$), reflecting an early stage in terms of the representation it has learned. With $10$ epochs, accuracy and F1 rises to $0.70$ and $0.69$. The trend continues and at the $20$-epoch checkpoint we observe (Acc $= 0.80$, F1 $= 0.79$). So, longer pre-training leads to more stable and useful representations, strongly improving downstream accuracy. We have updated the manuscript with the numbers from this experiment and are happy to expand this analysis to a few more downstream tasks in the final version.
> > We also observed that convergence during pre-training for **MnemoDyn** scales expectedly with dataset size: models trained on larger pre-training splits require more optimization steps to reach stability. However, it leads to a more stable fine-tuning phase.
> >
> > **5. Sensitivity to wavelet basis and number of scales**
> >
> > Yes, we had conducted several experiments to choose the wavelet basis where we explored different options from the Daubechies family of wavelet functions. The results indicated very minor variations when checking the impact of different wavelet families. So for simplicity, we simply stick with db2 for all experimental results presented in the paper.
> > Regarding the number of scales, we observed that increasing the number of scales led to faster computation as outlined in Section 2. We choose the number of scales such that the coarsest matrix being materialized had roughly $50 \times 50$ dimensions to give enough expressive power to the model. We mention that the number of scales is in general dependent on the sequence length of the original input signal. We are happy to include a simple ablation experiment for the choice of wavelet basis functions and the number of levels in the final version of the paper.
> >
> > **6. Clarity of methods section**
> >
> > We appreciate the reviewer’s comment about clarity and will revise the methods section to be more concise and intuitive. We have improved the schematic illustrating the full pipeline – from wavelet decomposition to operator evolution to low-rank representation  and make the text more succinct. We believe that this will improve the overall readability and make the final version more accessible to the readers.
> >
> > **7. Additional ablation study**
> >
> > A key practical advantage of **MnemoDyn**’s multi-scale and low-rank parameterization is that it avoids the compute bottleneck common in alternatives that rely on dense operators or fully learned spatiotemporal kernels. Dense operator learning scales quadratically with the number of regions and sequence length, and becomes infeasible for large rs-fMRI cohorts and longer sequences. The structured design in MnemoDyn yields operators whose evaluation and parameter updates scale near linearly in the number of coefficients, making full-dataset pre-training and downstream fine-tuning feasible without needing access to significant hardware cost. This compute efficiency is what allows MnemoDyn to be trained on ~$40K$ sequences and support inexpensive task-specific adaptation. Following the reviewer’s suggestion, we will include an ablation study detailing the effect of removing multi-scale structure and low-rank factorization in the final version. These additions will make the contribution clearer and help isolate the components driving performance.
> >
> > If the clarification(s) above addresses the reviewer’s concern, we will upload the updated manuscript promptly to reflect these revisions.

---

> > > ### Author Response · Authors · 2025-12-04
> > >
> > > We thank the reviewer for their time, thoughtful feedback and constructive suggestion(s) which have improved the presentation of our work. We have uploaded a revised pdf with the suggested and promised changes, marked in blue.

---

### Meta-Review · Area_Chair_DW7H · 2026-01-06

**Summary:**

MnemoDyn is a foundation model for resting-state fMRI that replaces transformer attention with an operator-based dynamical system over parcellated brain-region time series. It learns multiscale temporal dynamics using wavelet-parameterized kernels and a structured evolution/transition operator designed for sparse, block-diagonal computation, making it efficient on long sequences. The model is pretrained on ~40K rs-fMRI sequences pooled from many public and permissioned datasets spanning diverse populations and scanning protocols. In evaluations, MnemoDyn consistently improves rs-fMRI reconstruction quality over recent transformer-based and self-supervised baselines (e.g., BrainLM/Brain-JEPA) while using less compute. With lightweight fine-tuning heads, it also achieves strong performance on downstream tasks such as demographic/trait prediction and clinical classification (e.g., ADNI), including in small-sample regimes.

The authors provided an extensive rebuttal to the reviewers’ concerns, addressing many of the points raised. The revised manuscript, which incorporates the key clarifications, additional analyses, and methodological details from the rebuttal, is significantly stronger and should help prevent similar misunderstandings for future readers.

**Reviewer Concerns:**

Below are the main reviewers concerns and how the authors addressed them:

* Concern: reviewers asked for concrete runtime + memory benchmarks and scalability evidence.

  Response: authors provided single-GPU training time + peak memory for MnemoDyn and a rough comparison to Brain-JEPA’s reported multi-GPU training time; they also commit to adding a dedicated runtime/memory subsection in the final version.

* Concern: missing analysis of learned operator components (scales/frequencies/symbols/kernels) and interpretability of the transition matrix $A$.

  Response: authors added analyses decomposing  $A$ across multiscale components and reporting norm/sparsity patterns to show which parts dominate; they also added operator visualizations while explicitly avoiding over-claiming neurophysiological correspondence without validation.

* Concern: Reviewers felt the biological/neuroscientific motivation was overstated without validation/interpretability that maps to physiology.

  Response: Authors acknowledge this limitation explicitly, clarify MnemoDyn should not be read as a neuroscientific/physiological model, and revise text to calibrate claims; they position interpretability as model-level (temporal decomposition) rather than physiology.

* Concern: Table 1 lacked R² reconstruction metrics for key datasets, weakening comparisons.

  Response: Authors report adding both MSE and R² for UK Biobank and HCP, including cross-dataset reconstruction evaluations, and updating Table 1 accordingly.

* Concern: Trade-off between pretraining effort and downstream performance was asserted but not shown.

  Response: Authors added a checkpointing-style analysis showing downstream performance improving with pretraining epochs (reported for at least one downstream task during rebuttal) and plan to expand.

* Concern: Reviewer requested calibrated “foundation model” framing and quantitative scaling with model/data size.

   Response: Authors state they clarified scope as an rs-fMRI foundation model and added scaling results (model-size and dataset-size scaling tables), plus discussion of limits (e.g., cross-modal generalization).

**Reviewer Scores:**

The paper received three reviews: two marginally above the acceptance threshold and one reject. Interestingly, the authors’ rebuttal appears to have addressed the key concerns raised by the negative reviewer—as the reviewer noted—yet this improvement is not reflected in the final score.

Overall, I believe this is a strong paper, and the thorough rebuttal, together with the clarifications and added analyses, pushes it clearly above the acceptance threshold.

---

### Decision · Program_Chairs · 2026-01-26

Accept (Poster)